# Internet of Agents: Weaving a Web of Heterogeneous Agents for Collaborative Intelligence

**Weize Chen**[1*], **Ziming You**[2*], **Ran Li**[1*], **Yitong Guan**[2*], **Chen Qian**[1], **Chenyang Zhao**[1]
**Cheng Yang**[3], **Ruobing Xie**[4], **Zhiyuan Liu**[1✉], **Maosong Sun**[1]
[1] Tsinghua University, [2] Peking University
[3] Beijing University of Posts and Telecommunications, [4] Tencent
chenwz21@mails.tsinghua.edu.cn, ran.li572482@gmail.com
{zimingyou, 2101210206}@stu.pku.edu.cn, liuzy@tsinghua.edu.cn

## Abstract

The rapid advancement of large language models (LLMs) has paved the way for the development of highly capable autonomous agents. However, existing multi-agent frameworks often struggle with integrating diverse capable third-party agents due to reliance on agents defined within their own ecosystems. They also face challenges in simulating distributed environments, as most frameworks are limited to single-device setups. Furthermore, these frameworks often rely on hard-coded communication pipelines, limiting their adaptability to dynamic task requirements. Inspired by the concept of the Internet, we propose the Internet of Agents (IoA), a novel framework that addresses these limitations by providing a flexible and scalable platform for LLM-based multi-agent collaboration. IoA introduces an agent integration protocol, an instant-messaging-like architecture design, and dynamic mechanisms for agent teaming and conversation flow control. Through extensive experiments on general assistant tasks, embodied AI tasks, and retrieval-augmented generation benchmarks, we demonstrate that IoA consistently outperforms state-of-the-art baselines, showcasing its ability to facilitate effective collaboration among heterogeneous agents. IoA represents a step towards linking diverse agents in an Internet-like environment, where agents can seamlessly collaborate to achieve greater intelligence and capabilities[1].

## 1 Introduction

The Internet has revolutionized collaboration and knowledge-sharing, connecting people with diverse skills and backgrounds from around the world. This global network has enabled remarkable projects like Wikipedia and the development of Linux, achievements that would have been impossible for any single individual. By fostering collaboration, the Internet pushes the boundaries of human achievement, turning the once-impossible into reality.

The success of the Internet in enabling human collaboration raises an intriguing question: can we create a similar platform to facilitate collaboration among autonomous agents? With the rapid advancements in LLMs (OpenAI, 2023; Reid et al., 2024), we now have autonomous agents capable of achieving near-human performance on a wide range of tasks. These LLM-based agents have demonstrated the ability to break down complex tasks into executable steps, leverage various tools, and learn from feedback and experience (Qin et al., 2023; Wang et al., 2023c; Shinn et al., 2023; Qian et al., 2023b). As the capabilities of these agents continue to grow, and with an increasing number of third-party agents with diverse skills consistently emerging (Chase, 2022; Team, 2023; Significant Gravitas, 2023; Open Interpreter, 2023), it is crucial to explore how we can effectively and efficiently orchestrate their collaboration, just as the Internet has done for humans.

---

[*]Equal Contribution.    ✉ Corresponding author.
[1]https://github.com/OpenBMB/IoA

To address this challenge, we propose the concept of the Internet of Agents (IoA), a general framework for agent communication and collaboration inspired by the Internet. IoA aims to address three fundamental limitations of existing multi-agent systems (MAS) (Chen et al., 2023; Wu et al., 2023; Hong et al., 2023; Qian et al., 2023a): (1) **Ecosystem Isolation**: Most frameworks only consider agents defined within their own ecosystems, potentially blocking the integration of various third-party agents and limiting the diversity of agent capabilities and the platform's generality; (2) **Single-Device Simulation**: Nearly all MAS simulate MAS on a single device, which differs significantly from real-world scenarios where agents could be distributed across multiple devices located in different places; (3) **Rigid Communication and Coordination**: The communication process, agent grouping, and state transitions are mostly hard-coded, whereas in real life, humans decide on teammates based on the task at hand and dynamically switch between collaboration states.

To overcome these limitations, we propose an agent integration protocol that enables different third-party agents running on different devices to be seamlessly integrated into the framework and collaborate effectively. Additionally, we introduce an instant-messaging-app-like framework that facilitates agent discovery and dynamic teaming. By autonomously searching for potential agents capable of handling the tasks at hand, agents dynamically decide to form different teams and communicate within various group chats. Inspired by Speech Act Theory (Searle, 1969), and its application in conventional MAS (Finin et al., 1994; Labrou et al., 1999), within each group chat, we abstract out several conversation states and provide a flexible and general finite-state machine mechanism to support autonomous state transition for agents.

We demonstrate the effectiveness of IoA through extensive experiments and comparisons with state-of-the-art autonomous agents. By integrating AutoGPT (Significant Gravitas, 2023) and Open Interpreter (Open Interpreter, 2023), we show that IoA achieves a win rate of 66 to 76% in open-domain task evaluations when compared to these agents individually. Furthermore, with only a few basic ReAct agents integrated, IoA outperforms previous works on the GAIA benchmark (Mialon et al., 2023). In the retrieval-augmented generation (RAG) question-answering domain, our framework substantially surpasses existing methods, with a GPT-3.5-based implementation achieving performance close to or even exceeding GPT-4, and effectively surpassing previous MAS.

The impressive performance of IoA across various domains highlights the potential of this paradigm for autonomous agents. As smaller LLMs continue to advance (Mesnard et al., 2024; Hu et al., 2024; Abdin et al., 2024), running agents on personal computer or even mobile device is becoming increasingly feasible. This trend opens up new opportunities for deploying MAS in real-world scenarios, where agents can be distributed across multiple devices and collaborate to solve complex problems. We believe that by further exploring and refining the IoA paradigm, more sophisticated and adaptable MAS can be developed, ultimately pushing the boundaries of what autonomous agents can achieve in problem-solving and decision-making.

## 2 FRAMEWORK DESIGN AND KEY MECHANISMS OF IoA

### 2.1 OVERVIEW OF IoA

IoA is designed as an instant-messaging-app-like platform that enables seamless communication and collaboration among diverse autonomous agents. Inspired by the concept of Internet, IoA addresses three fundamental challenges in MAS (Chen et al., 2023; Wu et al., 2023; Qian et al., 2023a): **(1) Distributed agent collaboration**: Unlike traditional frameworks that simulate MAS on a single device, IoA supports agents distributed across multiple devices and locations. **(2) Dynamic and adaptive communication**: IoA implements mechanisms for autonomous team formation and conversation flow control, allowing agents to adapt their collaboration strategies based on task requirements and ongoing progress. **(3) Integration of heterogeneous agents**: IoA provides a flexible protocol for integrating various third-party agents, expanding the diversity of agent capabilities within the system. To better illustrate the difference between IoA and other MAS, see Appendix A for a comparative analysis of key aspects of different MAS.

At its core, IoA consists of two main components: the server and the client. The server acts as a central hub, managing agent registration, discovery, and message routing. It enables agents with varying capabilities to find each other and initiate communication. The client, on the other hand, serves as a wrapper for individual agents, providing them with the necessary communication func-

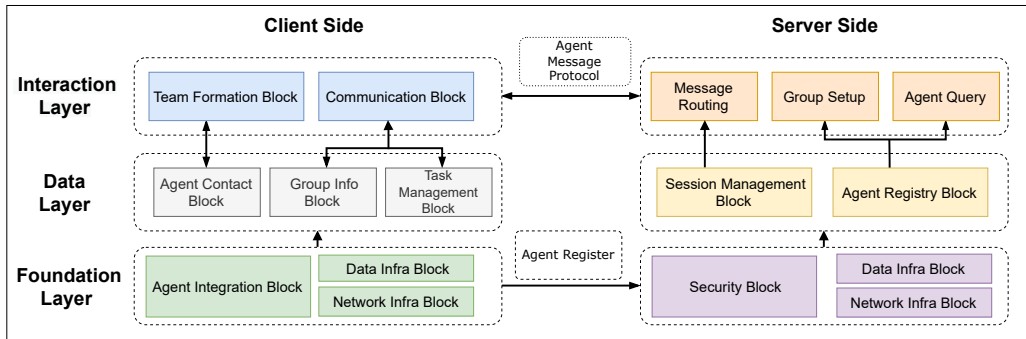

Figure 1: The illustration on the conceptual layered architecture on the design of IoA.

tionalities and adapting them to the specified protocol. IoA employs a layered architecture (Bass et al., 1999) for both the server and client components, comprising three layers:

- **Interaction Layer:** Facilitates team formation and agent communication.
- **Data Layer:** Manages information related to agents, group chats, and tasks.
- **Foundation Layer:** Provides essential infrastructure for agent integration, data management, and network communication.

These layers work together to facilitate agent collaboration through the network. In the following subsections, we will go through the IoA's architecture and design.

## 2.2 ARCHITECTURE OF IOA

The layered architecture of IoA is designed to support scalable, flexible, and efficient multi-agent collaboration. This architecture enables a clear separation of concerns and facilitates the integration of diverse agents and functionalities (Fig. 1).

### 2.2.1 SERVER ARCHITECTURE

The server acts as the central hub of IoA, facilitating agent discovery, group formation, and message routing. Its architecture consists of three layers. At the top level, the **Interaction Layer** manages high-level interactions between agents and the system. It encompasses the Agent Query Block for enabling agents to search for other agents based on specific characteristics, the Group Setup Block for facilitating the creation and management of group chats, and the Message Routing Block for ensuring efficient and accurate routing of messages between agents and group chats. The **Data Layer** Serves as the information backbone, it handles the storage and management of critical system information. The Agent Registry Block maintains a comprehensive database of registered agents, including their capabilities and current status, similar to service discovery in distributed systems (Meshkova et al., 2008; Netflix). Meanwhile, the Session Management Block manages active connections and ensures continuous communication between the server and connected clients. The **Foundation Layer** provides the essential infrastructure for the server's operations. It encompasses the Data Infrastructure Block for handling data persistence and retrieval, the Network Infrastructure Block for managing network communications, and the Security Block for implementing authentication, authorization, and other security measures to maintain system integrity.

### 2.2.2 CLIENT ARCHITECTURE

The client component of IoA serves as a wrapper for individual agents, providing them with the necessary interfaces to communicate within the system. Its architecture mirrors that of the server with three layers. The **Interaction Layer** manages the agent's interactions within the system. The Team Formation Block implements the logic for identifying suitable collaborators and forming teams for the task at hand, similar to coalition formation in conventional multi-agent research (Rahwan et al., 2009). Complementing this, the Communication Block manages the agent's participation in group chats and handles message processing. The **Data Layer** maintains local data relevant to the agent's operations. It includes the Agent Contact Block for storing information about other agents the current agent has interacted with, the Group Info Block for maintaining details about ongoing group

chats and collaborations, and the Task Management Block for tracking the status and progress of tasks assigned to the agent. The **Foundation Layer** provides the basic functionalities for the client's operations. The Agent Integration Block defines the protocols and interfaces for integrating third-party agents into the IoA ecosystem. Alongside this, the Data Infrastructure Block handles local data storage and retrieval, while the Network Infrastructure Block manages network communications with the server.

This layered architecture enables IoA to support a wide range of agent types and collaboration scenarios. By providing a clear separation of concerns and well-defined interfaces between layers, the architecture facilitates the integration of diverse agents and allows for future extensibility. Furthermore, this design supports the key mechanisms of IoA, as will be introduced in the next section.

## 2.3 KEY MECHANISMS

The effectiveness of IoA relies on several key mechanisms that enable seamless collaboration among diverse agents. These mechanisms work in concert to facilitate agent integration, team formation, task allocation, and structured communication. We detail these critical components in this section, with an example walk-through provided in Appendix B for better understanding.

### 2.3.1 AGENT REGISTRATION AND DISCOVERY

To enable collaboration among distributed agents with heterogeneous architectures, tools, and environments, we propose the agent registration and discovery mechanism. This mechanism forms the foundation for collaborative interactions within IoA, enabling the integration of diverse agents into the system and facilitating their discovery on the online server by other agents for potential collaboration through the network.

**Agent Registration:** When a new agent joins IoA, its client wrapper registers with the server by providing a comprehensive description of its capabilities, skills, and areas of expertise. This description, denoted as $d_i$ for an agent $c_i$, is stored in the Agent Registry Block of the server's Data Layer. Formally, we represent the set of all registered agents as $\mathcal{C} = \{c_1, c_2, ..., c_n\}$, where each $c_i$ is associated with its description $d_i$.

**Agent Discovery:** The agent discovery function leverages the information stored in the Agent Registry from the online server to enable agents to find suitable collaborators for specific tasks. Agents can use the `search_client` tool provided by the server's Agent Query Block to search for other agents based on desired characteristics or capabilities. Formally, let $\mathcal{L}_d = [l_1, l_2, ..., l_k]$ be a list of desired characteristics generated by an agent seeking collaborators. The `search_client` function can be represented as: `search_client` $: \mathcal{L}_d \rightarrow \mathcal{P}(\mathcal{C})$, where $\mathcal{P}(\mathcal{C})$ denotes the power set of $\mathcal{C}$. The function returns a subset of clients $\mathcal{C}_d \subseteq \mathcal{C}$ whose descriptions $d_j$ match the desired characteristics in $\mathcal{L}_d$. The matching process between $\mathcal{L}_d$ and $d_j$ can be implemented with various semantic matching techniques (Robertson & Zaragoza, 2009; Karpukhin et al., 2020). It ensures that agents with relevant capabilities can be discovered even if their descriptions do not exactly match the query.

### 2.3.2 AUTONOMOUS NESTED TEAM FORMATION

The autonomous nested team formation mechanism enables dynamic and flexible combinations of agents based on task requirements, allowing for the creation of nested sub-teams for complex tasks.

**Team Formation Process:** When a client $c_i \in \mathcal{C}$ is assigned a task $t$, it initiates the team formation process using two essential tools provided by the server: `search_client` and `launch_group_chat`. The client's LLM decides which tool to call based on the task and the current set of discovered clients. If more collaborators are needed, it calls `search_client` with appropriate characteristics. Once suitable collaborators are found, it calls `launch_group_chat` to initiate a new group chat $g \in \mathcal{G}$, where $\mathcal{G}$ is the space of all group chats.

**Nested Team Structure:** The nested team formation allows for a hierarchical structure of teams and sub-teams. Let $g_0 \in \mathcal{G}$ be the initial group chat for task $t$. During the execution of $t$, if a client $c_i$ is assigned with a sub-task $t_l$ (the task assignment mechanism will be introduced in Section 2.3.4), and it identifies $t_l$ requires additional expertise, $c_i$ is allowed to search for appropriate agents again and initiate a new sub-group chat $g_l \in \mathcal{G}$. This process can continue recursively,

forming a tree-like structure of group chats. Formally, we can define a function $h : \mathcal{G} \rightarrow \mathcal{P}(\mathcal{G})$ that maps a group chat to its set of sub-group chats. The nested structure can be represented as: $h(g) = \{g_1, g_2, ..., g_m\}, \quad h(g_i) = \{g_{i1}, g_{i2}, ..., g_{in}\}$, and so on.

**Communication Complexity:** The nested team formation mechanism can theoretically reduce the communication complexity in large agent teams in terms of communication channels. Assume that the minimal agent setting capable of completing a given task is $g$ with $|g|$ members. Consider two types of team structures: a fully connected team structure and a nested team structure ($h(g) = \{g_1, g_2, ..., g_m\}$ where $\bigcup_{i=1}^{m} g_i = g$. In the first scenario, communication channels (connected edges) are $C_{\text{full}}(g) = \{(g_i, g_j) | 0 \leq i, j \leq m\}$ with $c_{\text{full}}(g) = |C_{\text{full}}(g)| = \frac{|g|(|g|-1)}{2}$. For the Nested Team Structure, it necessarily follows that $c_{\text{nested}}(g) = |\bigcup_{g_i \in h(g)} C_{\text{full}}(g_i)| \leq c_{\text{full}}(g)$. Fig. 2 illustrates an example of the nested team formation pro-

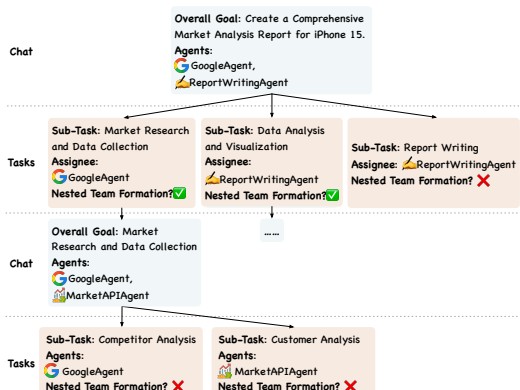

Figure 2: An example of nested team formation mechanism. The process is simplified for clarity.

cess. In this example, the initial group chat $g_0$ spawns three sub-group chats $g_1$, $g_2$ and $g_3$ for specific sub-tasks during the discussion. $g_1$ further creates two sub-group chats $g_{21}$ and $g_{22}$ for a more specialized sub-task.

### 2.3.3 AUTONOMOUS CONVERSATION FLOW CONTROL

Effective communication is crucial for successful collaboration among autonomous agents. Inspired by Speech Act Theory (Austin, 1975; Searle, 1969) and its applications in conventional MAS (Finin et al., 1994; Labrou et al., 1999), we introduce an autonomous conversation flow control mechanism in IoA. This mechanism enables agents to coordinate their communication and maintain a structured dialogue, enhancing the efficiency and effectiveness of their collaboration.

**Sequential Speaking Mechanism:** To manage potential conflicts and ensure clear communication, IoA adopts a sequential speaking mechanism where only one agent is permitted to speak at a time. This approach, while simple, provides a foundation for more sophisticated conversation management when combined with the following dynamic features.

**Finite State Machine for Group Chat States:** We formalize the conversation flow as a finite state machine $M = (S, \Sigma, \delta, s_0, F)$, where:

- $S = \{s_d, s_s, s_a, s_p, s_c\}$ is the set of states representing discussion, synchronous task assignment, asynchronous task assignment, pause & trigger, and conclusion, respectively.
- $\Sigma$ is the state transition decision space.
- $\delta : S \times \Sigma \rightarrow S$ is the transition function mapping the current state and the transition decision made by LLMs to the next state.
- $s_0 = s_d$ is the initial state, representing the start of the conversation in the discussion phase.
- $F = \{s_c\}$ is the set of final states, containing only the conclusion state.

Figure 3 illustrates the state transitions in the conversation flow. Each state corresponds to different phases of the collaboration process: (1) *Discussion* ($s_d$): Agents engage in general dialogue, exchange ideas, and clarify task requirements. (2) *Synchronous task assignment* ($s_s$): Tasks are assigned to specific agents, pausing the group chat until completion (Section 2.3.4). (3) *Asynchronous task assignment* ($s_a$): Tasks are assigned without interrupting the ongoing discussion (Section 2.3.4). (4) *Pause & trigger* ($s_p$): The group chat is paused, waiting for the completion of specified asynchronous tasks. (5) *Conclusion* ($s_c$): Marks the end of the collaboration, prompting a final summary. These states align with

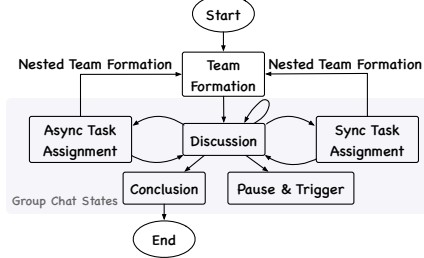

Figure 3: The state transition diagram.

the speech acts in Speech Act Theory, e.g., assertives (discussion), directives (task assignment), commissives (pause & trigger), and declarations (conclusion) (Searle, 1976).

**Autonomous State Transitions and Next Speaker Selection:** Recent studies have demonstrated the efficacy of LLMs in autonomously managing state transitions within predefined state spaces (Liu & Shuai, 2023; Wu et al., 2024a), with state machines often enhancing overall system performance (Li et al., 2024). In IoA, the LLM within each client determines state transitions and selects the subsequent speaker. Let $\mathcal{M}_t$ be the set of messages exchanged up to time step $t$. We define the decision function of the LLM as: $f_{\text{LLM}} : \mathcal{M}_t \times S \rightarrow S \times \mathcal{C}$, where $S$ is the set of states and $\mathcal{C}$ is the set of clients. The next state $s_{t+1}$ and the next speaker $c_{t+1}$ are determined as: $(s_{t+1}, c_{t+1}) = f_{\text{LLM}}(\mathcal{M}_t, s_t)$. This decision-making process considers factors such as the completion of assigned tasks, the need for further discussion, and the overall goals of the collaboration. The autonomous selection of the next speaker ensures that the most relevant agents are involved at appropriate times, promoting efficient information exchange and problem-solving.

By implementing this autonomous conversation flow control mechanism, IoA enables structured and efficient communication among agents, facilitating more effective problem-solving and decision-making in complex multi-agent scenarios.

### 2.3.4 TASK ASSIGNMENT AND EXECUTION

The task assignment and execution mechanism in IoA is designed to distribute work efficiently among agents and manage the execution of both simple and complex tasks. This mechanism works in concert with the team formation and conversation flow control mechanisms to ensure effective collaboration and task completion.

**Task Representation:** In IoA, a task $t \in \mathcal{T}$ is represented as a tuple $(d_t, \mathcal{S}_t)$, where $d_t$ is the task description and $\mathcal{S}_t = \{s_1, s_2, ..., s_n\}$ is the set of sub-tasks that $t$ can be decomposed into. Initially, $\mathcal{S}_t$ may be empty, with sub-tasks being identified dynamically during the collaboration process.

**Task Allocation:** Task allocation in IoA occurs within the context of group chats and is closely tied to the conversation flow control mechanism. There are two types of task allocation:

1. *Synchronous Task Allocation:* When the group chat enters the synchronous task assignment state $s_s$, tasks are allocated to specific agents, and the group chat is paused until the tasks are completed.
2. *Asynchronous Task Allocation:* In the asynchronous task assignment state $s_a$, tasks are allocated without interrupting the ongoing discussion. This allows for parallel execution of tasks.

Formally, we can define a task allocation function $\alpha : \mathcal{T} \times \mathcal{G} \rightarrow \mathcal{P}(\mathcal{C})$, which maps a task and a group chat to a subset of clients responsible for executing the task.

**Task Execution:** Once a task is allocated, the responsible agent(s) begin execution. For integrated third-party agents, task execution is handled through the Agent Integration Block in the client's Foundation Layer. This block provides a standardized interface for task execution, typically in the form: `run : String → TaskID`, where the input is the task description, and the output is a unique identifier for the task.

Upon completion of a task or sub-task, the responsible agent(s) report back to the group chat. In the case of synchronous tasks, this triggers the resumption of the group chat. For asynchronous tasks, the completion is noted, and any relevant information is shared with the group.

The pause & trigger state $s_p$ in the conversation flow control mechanism plays a crucial role in managing the completion of multiple asynchronous tasks. It allows the group chat to wait for the completion of specified asynchronous tasks before proceeding, ensuring that all necessary information is available for subsequent stages of the collaboration.

## 3 EXPERIMENTS

To demonstrate the effectiveness and versatility of IoA in integrating heterogeneous agents, we conduct comprehensive experiments across diverse tasks. These experiments are designed to showcase different aspects of agent heterogeneity: tool variability (Section 3.1), architectural diversity

Table 1: The performance on the validation set of GAIA benchmark.

| Models | Agent Type | Level 1 | Level 2 | Level 3 | Overall |
|---|---|---|---|---|---|
| GPT-4 | 👤 | 15.09 | 2.33 | 0.00 | 6.06 |
| GPT-4-Turbo | 👤 | 20.75 | 5.81 | 0.00 | 9.70 |
| AutoGPT-4 (Significant Gravitas, 2023) | 👤 | 13.21 | 0.00 | 3.85 | 4.85 |
| GPT-4 + Plugins (Mialon et al., 2023) | 👤 | 30.30 | 9.70 | 0.00 | 14.60 |
| FRIDAY (Wu et al., 2024b) | 👤 | 45.28 | 34.88 | 11.54 | 34.55 |
| AutoGen (Wu et al., 2023) | 👥 | **54.72** | 38.37 | 11.54 | 39.39 |
| IoA | 👥 | 50.94 | **40.70** | **15.38** | **40.00** |

(Section 3.2), disparate observation and action spaces (Section 3.3), and varied knowledge bases (Section 3.4). Due to space limit, we place the analysis on the average cost, nested team formation precision and communication case study in Appendices D and E. Our objective is twofold: (1) to illustrate IoA's proficiency in facilitating collaboration among heterogeneous agents, and (2) to highlight its adaptability across various problem domains. We present our experimental results and offer comparative analyses between IoA and state-of-the-art (SoTA) approaches for each task category. If not specified, we use GPT-4-1106-preview model in our experiments. The prompts within IoA are kept the *same* across different tasks, and are not specifically tuned for a certain task.

## 3.1 HETEROGENEOUS TOOLS: GAIA BENCHMARK

To evaluate IoA's capability in integrating agents with heterogeneous tools, we employ the GAIA benchmark (Mialon et al., 2023). This benchmark comprises a diverse set of real-world questions designed to assess an agent system's proficiency in solving complex tasks through the synergistic application of multiple skills, including natural language understanding, reasoning, and external knowledge integration. The benchmark's three-tiered difficulty structure provides a robust testbed for evaluating the capability of agent systems.

**Experimental Setups:** We instantiate IoA with four basic ReAct agents (Yao et al., 2023), each equipped with a distinct tool: a web browser, a code interpreter, a Wikidata searcher, and a YouTube video transcript downloader. This configuration allows us to assess IoA's ability to orchestrate collaboration among agents with heterogeneous tools. We benchmark IoA against several SoTA agent systems, evaluating performance across all three difficulty levels of GAIA. Detailed implementation specifics are provided in Appendix F.4.1.

**Results and Analysis:** The results, presented in Table 1, demonstrate IoA's superior performance across the GAIA benchmark. Despite utilizing only basic ReAct agents, IoA achieves the highest overall performance, surpassing all other approaches. Notably, IoA excels in the more challenging Level 2 and Level 3 tasks, which require advanced reasoning and intricate collaboration.

Compared to AutoGen, IoA demonstrates superior performance in two out of three difficulty levels. This superiority can be attributed to IoA's collaboration mechanisms and the flexibility of integrating agents with different tools, while in AutoGen, only one agent utilizes different tools, and other agents act as feedback providers. IoA enables adaptive team composition and efficient sub-task execution, leading to enhanced performance on complex, multi-faceted problems.

These results highlight IoA's effectiveness as an orchestrator for diverse agents in solving real-world, multi-step problems. By providing a flexible and efficient platform for agent collaboration, IoA enables even basic agents to achieve SoTA performance, outperforming more sophisticated standalone agents and representative MAS.

## 3.2 HETEROGENEOUS ARCHITECTURE: OPEN-ENDED INSTRUCTION BENCHMARK

To evaluate IoA's capability in integrating and orchestrating agents with heterogeneous architectures, we develop a comprehensive benchmark comprising 153 open-ended instructions with self-instruct (Wang et al., 2023e). This benchmark spans four diverse categories: search & report, coding, mathematics, and life assistance. Unlike the GAIA benchmark, which primarily focuses on question-answering tasks with deterministic answers, our curated benchmark incorporates a higher proportion of non-QA tasks requiring generative responses. This design choice aims to better reflect

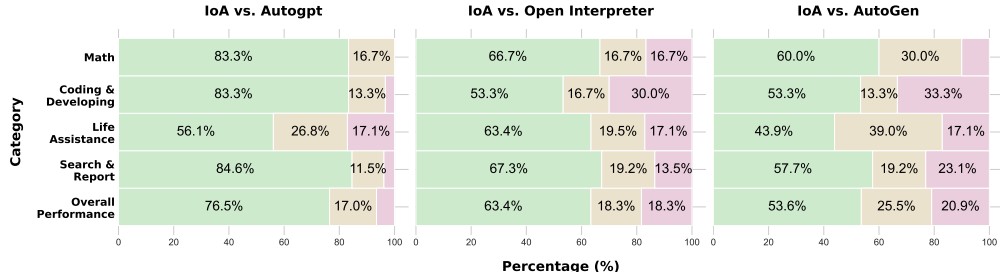

Figure 4: Comparison of win rates on the open-ended instruction benchmark between IoA, Auto-GPT, Open Interpreter and AutoGen. Green : win rate; yellow : tie rate; red : loss rate.

Table 2: Average success rate and the number of steps on different tasks from RoCoBench.

| Model | Metric | Cabinet | Sweep | Sandwich | Sort | Rope |
|---|---|---|---|---|---|---|
| Central Plan (oracle) | Success | 0.90 | **1.00** | 0.96 | 0.70 | 0.50 |
| | #Step | 4.0 | 8.4 | 8.8 | 8.6 | 2.3 |
| Roco Dialog | Success | 0.75 | 0.70 | 0.70 | 0.70 | **0.70** |
| | #Step | 4.7 | 7.9 | 9.1 | 5.4 | 2.4 |
| IoA | Success | **1.00** | 0.80 | **1.00** | **1.00** | **0.70** |
| | #Step | 4.6 | 8.5 | 8.9 | 5.8 | 2.6 |

the diverse nature of real-world challenges that agent systems are expected to address. The curation process is elaborated at Appendix F.4.2.

**Experimental Setups:** We integrate two SoTA third-party agents with distinct architectures: AutoGPT (Significant Gravitas, 2023) and Open Interpreter (Open Interpreter, 2023), into IoA. The integration process, detailed in Appendix F.4.2, demonstrates IoA's versatility in accommodating agents with divergent internal structures and operational paradigms. This configuration allows us to assess IoA's efficacy in facilitating collaboration among independently developed agents with heterogeneous architectures.

For evaluation, we employ GPT-4-1106-preview as an impartial judge, a choice supported by previous research demonstrating high agreement between GPT models and human evaluators in assessing response quality (Chiang et al., 2023; Zheng et al., 2023a; Chan et al., 2023). To mitigate potential order-induced biases, we implement a robust evaluation approach following Zheng et al. (2023a), where the order of responses is alternated in the prompt. A "win" is only declared when one competitor is consistently judged superior across both orderings.

**Results and Analysis:** As shown in Fig. 4, IoA consistently outperforms both individual agents across all task categories, and is also generally better than AutoGen. Overall, IoA achieves a remarkable win rate of 76.5% against AutoGPT and 63.4% against Open Interpreter. These results underscore IoA's proficiency in efficiently gathering and synthesizing information, as well as its effectiveness in facilitating collaborative problem-solving across diverse domains. Additionally, we demonstrate in Appendix C that fine-tuning a Llama 3 8B (Meta, 2024) on the communication trajectories collected from various tasks enables it to serve as the communication layer LLM, outperforming standalone agents and remaining comparable to AutoGen powered by GPT-4.

The demonstrated capability of IoA to seamlessly integrate and orchestrate agents with heterogeneous architectures enables the harness of the strengths of diverse, independently developed agents, making it possible to create more versatile and capable agent systems. Recent studies on scaling laws in MAS (Qian et al., 2024) indicate that MAS performance improves as the number of agents increases. By design, IoA provides a robust foundation for integrating a greater variety of agents, potentially leading to better scaling laws.

### 3.3 HETEROGENEOUS OBSERVATION AND ACTION SPACE: EMBODIED AGENT TASKS

To evaluate IoA's efficacy in orchestrating agents with heterogeneous observation and action spaces, we conduct experiments using RoCoBench (Mandi et al., 2023), a benchmark for assessing collaboration and communication capabilities of embodied agents. RoCoBench comprises six collaborative

tasks requiring two or three agents with partial, often distinct action spaces or observations to cooperate towards a common objective.

**Experimental Setups:** We benchmark IoA against two baselines established by Mandi et al. (2023): (1) Central Plan, a centralized agent with complete environmental information and control, and (2) Roco Dialog, a specialized MAS designed for this task. Given that RoCoBench requires agents to output action plans in a specific format rather than interact with tools, we adapt IoA to this scenario without integrating external agents. Instead, we provide environmental observations to two IoA clients and extract their action plans from their discussion. This setup allows us to evaluate IoA's ability to manage agents with heterogeneous observation and action spaces. Detailed implementation specifics are available in Appendix F.4.3. To ensure a fair comparison, we conduct 10 runs for both IoA and Roco Dialog for each task, reporting average success rates and steps taken. Results for Central Plan are sourced directly from Mandi et al. (2023). Note that the Pack Grocery task is omitted due to implementation errors in the benchmark release.

**Results and Analysis:** As shown in Table 2, IoA outperforms Roco Dialog in four out of five tasks in terms of success rate, despite not being specifically optimized for embodied tasks. IoA achieves perfect scores on the Cabinet, Sandwich, and Sort tasks, demonstrating the robustness of its communication and collaboration mechanisms in enabling embodied agents with heterogeneous observation and action spaces to work synergistically towards common goals. Notably, IoA's success rates are superior or comparable to Central Plan across tasks, although it generally requires slightly more decision steps for task completion. Given that IoA is a general multi-agent framework not specifically designed for embodied AI tasks, the marginal increase in step count is a reasonable trade-off for its versatility and effectiveness.

### 3.4 HETEROGENEOUS KNOWLEDGE: RETRIEVAL-AUGMENTED GENERATION

To evaluate IoA's efficacy in orchestrating agents with heterogeneous knowledge, we conduct experiments on retrieval-augmented generation (RAG) tasks (Lewis et al., 2021). RAG tasks present a unique challenge where agents must retrieve relevant information from diverse sources and collaborate to synthesize accurate responses, making them an ideal testbed for assessing IoA's ability to manage knowledge heterogeneity and facilitate effective inter-agent communication.

**Experimental Setups:** We implement IoA with GPT-3.5-turbo-0125 as the core language model, following Apollo's Oracle (Wang et al., 2023b). To evaluate knowledge heterogeneity and its impact, we design three scenarios: (1) *Heterogeneous Knowledge*: Two clients access different evidence pools (Wikipedia/Google), testing IoA's ability to manage knowledge heterogeneity. (2) *Homogeneous Knowledge (2 Agents)*: Two clients access both pools, serving as a control to isolate heterogeneity effects. (3) *Homogeneous Knowledge (3 Agents)*: Three clients access both pools, assessing scalability and knowledge redundancy trade-offs.

This design allows us to disentangle the effects of knowledge heterogeneity from agent count and knowledge redundancy. We evaluate across four datasets: TriviaQA (Joshi et al., 2017), Natural Questions (NQ) (Kwiatkowski et al., 2019), HotpotQA (Yang et al., 2018), and 2WikiMultiHopQA (2WMHQA) (Ho et al., 2020), using 250 randomly sampled question-answer pairs from each. Implementation details are in Appendix F.4.4.

**Results and Analysis:** Table 3 demonstrates IoA's remarkable performance across all datasets, often surpassing or matching GPT-4 despite being based on GPT-3.5. On two out of four tasks, IoA's heterogeneous knowledge scenario outperforms homogeneous Apollo's Oracle, showcasing IoA's effectiveness in managing knowledge diversity. This configuration achieves the best performance on NQ and competitive results on other datasets, often outperforming single-model approaches and specialized frameworks like Apollo's Oracle.

We also conduct experiments in homogeneous settings. IoA with 3 agents achieves the best overall performance, outperforming all baselines on TriviaQA and showing competitive results on other datasets. Interestingly, the 2-agent homogeneous scenario outperforms the 3-agent setup on HotpotQA and 2WMHQA, suggesting that optimal agent configuration may be task-dependent. These results not only validate IoA's effectiveness in RAG tasks but also highlight its potential as a versatile platform for managing both heterogeneous and homogeneous knowledge in MAS.

| Model | TriviaQA | NQ | HotpotQA | 2WMHQA | Overall |
|---|---|---|---|---|---|
| GPT 4 | 0.902 | 0.692 | 0.566 | 0.284 | 0.611 |
| GPT 3.5 Turbo | 0.778 | 0.532 | 0.384 | 0.210 | 0.476 |
| + Zero-Shot CoT (Wei et al., 2022) | 0.772 | 0.588 | 0.410 | 0.190 | 0.490 |
| + Self Consistency (Wang et al., 2023d) | 0.818 | 0.622 | 0.408 | 0.206 | 0.514 |
| + Reflxion (Shinn et al., 2023) | 0.762 | 0.586 | 0.378 | 0.254 | 0.495 |
| + Multi-Agent Debate1 (Du et al., 2023) | 0.798 | 0.648 | 0.394 | 0.186 | 0.507 |
| + Multi-Agent Debate2 (Liang et al., 2023) | 0.756 | 0.576 | 0.450 | 0.334 | 0.529 |
| Apollo's Oracle (Homogeneous) | 0.834 | 0.662 | 0.542 | 0.350 | 0.597 |
| IoA + 2 Agents (Heterogeneous) | 0.803 | **0.708** | 0.478 | 0.449 | 0.610 |
| IoA + 2 Agents (Homogeneous) | 0.820 | 0.671 | **0.586** | **0.530** | 0.652 |
| IoA + 3 Agents (Homogeneous) | **0.908** | 0.682 | 0.575 | 0.519 | **0.671** |

Table 3: Model Accuracy for RAG task. IoA (GPT-3.5) matches or exceeds GPT-4 across all tasks. Best results are in bold; second-best (excluding GPT-4) are underlined. *Heterogeneous* refers to agents with different evidence pools, while *Homogeneous* means all agents share all evidence pools.

# 4    RELATED WORK

**LLM-based Agents**    Recent advancements in LLMs, such as GPT (OpenAI, 2023), Claude (Anthropic, 2024) and Gemini (Reid et al., 2024), have led to the development of highly capable AI agents, which can engage in natural language interactions and perform a wide range of tasks. To enhance the capabilities of LLM-based agents, researchers have explored the integration of external tools and knowledge sources (Nakano et al., 2021; Yao et al., 2023; Schick et al., 2023; Shen et al., 2023), enabling agents to access and utilize relevant information beyond their pre-trained knowledge. The various agents have demonstrated significant progress in a wide range of domains, including operating system interactions, software engineering, and general AI applications. For instance, OS-Copilot facilitates generalist interactions across web browsers and code terminals (Wu et al., 2024b), while OpenDevin focuses on autonomous software development tasks such as coding and debugging (OpenDevin Team, 2024). Other notable developments include XAgent for complex task solving (Team, 2023) and Voyager (Wang et al., 2023a), an open-ended embodied agent leveraging LLMs for Minecraft game-playing. These advancements have laid the foundation for more sophisticated and versatile LLM-based agents, capable of autonomous task execution.

**LLM-based Multi-Agent Systems**    Building upon the success of individual LLM-based agents, researchers have begun to explore the potential of multi-agent systems composed of these agents. Early works demonstrated the feasibility of using LLMs to simulate multi-agent interactions and emergent behaviors (Park et al., 2023). Since then, various approaches have been proposed to enable effective collaboration and communication among LLM-based agents. Frameworks such as AgentVerse (Chen et al., 2023) and AutoGen (Wu et al., 2023) provide the necessary infrastructure for agent collaboration. In software development, multi-agent systems like ChatDev (Qian et al., 2023a), MetaGPT (Hong et al., 2023) have shown promising results in automating coding, testing, and debugging processes. Despite these advancements, significant limitations remain, such as the lack of support for integrating diverse third-party agents, the inability to support distributed multi-agent systems, and the reliance on hard-coded communication protocols and state transitions. IoA aims to address these limitations and provide a more flexible and scalable platform for LLM-based multi-agent collaboration, paving the way for more advanced and practical systems that can tackle complex real-world problems effectively.

# 5    CONCLUSION

We introduced IoA, a novel LLM-based multi-agent collaboration framework inspired by the Internet. It overcomes limitations of existing frameworks by offering a flexible, scalable platform for integrating diverse third-party agents, enabling distributed collaboration, and introducing dynamic teaming and conversation control. Extensive experiments on various benchmarks demonstrate IoA's effectiveness in facilitating heterogeneous agent collaboration, consistently outperforming SoTA baselines. We believe IoA will serve as a foundation for future research, enabling the integration of independently developed agents and advancing multi-agent systems.

ACKNOWLEDGMENT

This work was supported by the National Key R&D Program of China (No.2022ZD0116312), the Postdoctoral Fellowship Program of CPSF under Grant Number GZB20230348, the Young Elite Scientists Sponsorship Program by CAST (2023QNRC001) and Tencent Rhino-Bird Focused Research Program.

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

Table 4: Comparison of Key Features Across Different Frameworks. Candidates should be pre-specified in the team expansion from AutoGen.

| Key Features | IoA | AutoGen | MetaGPT | AgentVerse | Crew AI | LangGraph | CAMEL |
|---|---|---|---|---|---|---|---|
| Heterogeneity | ✓ | ✗ | ✗ | ✗ | ✗ | ✗ | ✗ |
| Distributed | ✓ | ✗ | ✗ | ✗ | ✗ | ✗ | ✗ |
| Auto Teaming | ✓ | ✗ | ✗ | ✓ | ✗ | ✗ | ✗ |
| Team Expansion | ✓ | ✓* | ✗ | ✗ | ✗ | ✗ | ✗ |
| Async Tasks | ✓ | ✗ | ✓ | ✗ | ✓ | ✗ | ✗ |
| Flow Control | Autonomous | Pre-Defined | Pre-Defined | Pre-Defined | Pre-Defined | Supervisor-Managed | Turn-Taking |

## A  COMPARATIVE ANALYSIS OF MULTI-AGENT SYSTEM FRAMEWORKS

To contextualize IoA's capabilities within the broader landscape of multi-agent systems, including AutoGen (Wu et al., 2023), AgentVerse (Chen et al., 2023), MetaGPT (Hong et al., 2023), Crew AI (CrewAIInc, 2024), LangGraph (AI, 2023) and CAMEL (Li et al., 2023), we conducted a comparative analysis of key features across different frameworks, as summarized in Table 4. This analysis reveals IoA's unique position in supporting agent heterogeneity and distributed collaboration, features not fully realized in other prominent frameworks. While some frameworks like AgentVerse and AutoGen offer partial support for autonomous team formation and dynamic expansion respectively, IoA integrates these capabilities more comprehensively, allowing for flexible team structures without pre-specified constraints.

IoA's autonomous conversation flow control further distinguishes it from other frameworks that rely on pre-defined, user-managed, or turn-taking approaches. This autonomy, combined with support for asynchronous task execution (a feature shared with MetaGPT and Crew AI), enables IoA to handle complex, multi-faceted projects with greater adaptability. The unique combination of these features in IoA —heterogeneity, distributed collaboration, autonomous teaming and flow control, and asynchronous task handling—positions it as a versatile solution for diverse multi-agent collaboration scenarios.

## B  PUTTING IT ALL TOGETHER: A WALKTHROUGH OF IoA IN ACTION

To illustrate the integrated functionality of IoA, in Fig. 5, we present an example walkthrough of the system with an illustrative complex task: writing a research paper on the Internet of Agents. Initially, client $c_1$, an AI research specialist trained additionally on AI academic paper, engages the Team Formation Block, utilizing the `search_client` function with a list of keywords {Internet, Multi-Agent System Specialist, Paper Writing, LLM Expert}. The server returns a set of matched clients $\{c_2, c_3, c_4, c_5\}$, from which $c_1$ forms group $g_0$ with members $\{c_1, c_2, c_3\}$ via `launch_group_chat`, where $c_2$ has access to scholarly databases and $c_3$ specializes in academic writing.

Upon the formation of group chat $g_0$, all clients transition to the Communication Block for $g_0$, where the autonomous conversation flow control mechanism, implemented as a finite state machine, guides the collaboration. The process begins with brainstorming in the discussion state ($s_d$), progressing to task assignment states ($s_s, s_a$) where agents are allocated specific responsibilities. For instance, $c_2$ is tasked with conducting a literature review using its access to scholarly resources. The nested team formation mechanism is demonstrated when $c_2$ identifies a need for specialized PDF expertise. This prompts $c_2$ to initiate a sub-group formation process, resulting in the creation of sub-group $g_1$ with a new agent $c_6$, a PDF expert. Throughout the process, the conversation alternates between discussion ($s_d$) and asynchronous task assignment ($s_a$) states, facilitating parallel work on assigned tasks. The message protocol ensures efficient communication, enabling the exchange of ideas, citations, and draft segments across the nested group structure.

In the final integration phase, the group enters a synchronous task assignment state ($s_s$) for collaborative editing and refinement, demonstrating IoA's capacity for coordinating intensive, real-time collaboration among multiple agents. The process concludes with a transition to the conclusion state ($s_c$), where a final review is conducted and the paper is prepared for submission.

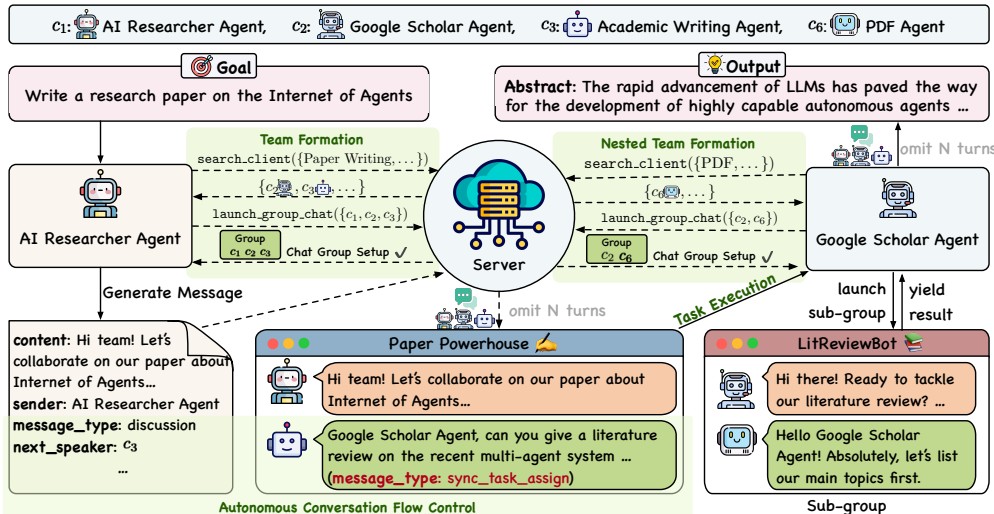

Figure 5: An example walkthrough of the major components of IoA.

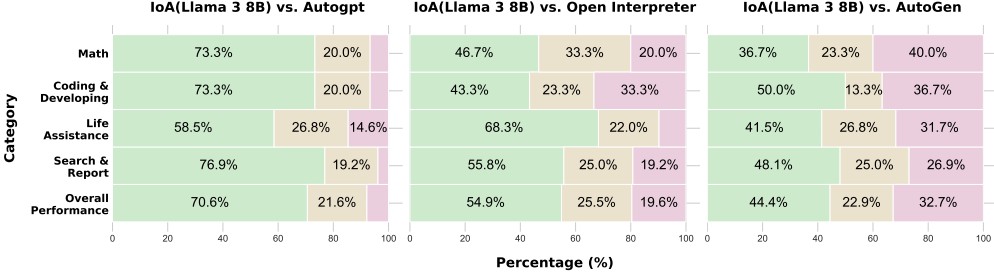

Figure 6: Comparison of win rates on the open-ended instruction benchmark between IoA, Auto-GPT, Open Interpreter and AutoGen.

## C LLAMA AS THE COMMUNICATION LLM IN OPEN-ENDED INSTRUCTION BENCHMARK

While we have shown that GPT-4 can have remarkable performance in facilitating agent communication when they are used as the communication LLM, their deployment can be expensive and may raise privacy concerns. By fine-tuning a smaller, locally deployable model to be communication LLM, we aim to create a more efficient and potentially more tailored communication layer for IoA. To explore the potential of using locally deployable models for the communication layer in IoA, we conducted an experiment involving fine-tuning Llama 3 8B (Meta, 2024) on the communication trajectories collected from various tasks presented in our paper.

### C.1 FINE-TUNING PROCESS

We fine-tuned the Llama 3 8B model using the collected communication trajectories. Since IoA heavily relies on JSON format output, we fine-tune the model to generate responses following the specification of IoA. We directly use the training recipes[2] provided by the alignment handbook (Tunstall et al.), and only change the model and data related hyper-parameters, and leave others unchanged.

---

[2] https://github.com/huggingface/alignment-handbook/blob/main/recipes/zephyr-7b-beta/sft/config_full.yaml

## C.2 EXPERIMENTAL SETUP

After fine-tuning, we deployed the Llama 3 8B model as the communication LLM in IoA[3]. We then evaluated its performance on the open-ended instruction benchmark, comparing it to: (1) Standalone Open Interpreter (2) Standalone AutoGPT (3) AutoGen, all the baselines are powered by GPT-4.

## C.3 RESULTS

The results presented in Fig. 6 demonstrate that IoA, utilizing a fine-tuned Llama 3 8B, consistently outperforms standalone agents such as AutoGPT and Open Interpreter across all task categories. Specifically, IoA achieves a notable overall win rate of 70.6% against AutoGPT and 54.9% against Open Interpreter. Furthermore, when compared to AutoGen, IoA remains competitive, illustrating that fine-tuned smaller models can serve as highly effective communication agents within IoA, enhancing its practicality.

These findings underscore the effectiveness of fine-tuning Llama 3 8B for communication in multi-agent coordination and show that a fine-tuned smaller model can successfully function as the communication LLM, effectively coordinating diverse agents within the system. IoA, incorporating a smaller model, either matches or surpasses other agent systems in most tasks, suggesting the potential for more efficient and locally deployable solutions.

## D  TEAM FORMATION PRECISION

To evaluate the precision of IoA's autonomous team formation mechanism, we developed a benchmark using GPT-4, comprising 625 diverse tasks paired with 1500 dummy agent profiles. This simulated environment allows us to assess the accuracy of both regular and nested team formation in a large-scale setting. Detailed data construction processes are available in Appendix H.

**Experimental Design:** We evaluate two distinct scenarios: regular team formation and nested team formation. For regular team formation, each task is associated with 2 or more suitable agent profiles generated by GPT. For nested team formation, we generate a subtask for each original task that can or cannot be completed by the initially formed team, if not, an additional agent profile capable of addressing this subtask is generated. We evaluate whether the team can correctly decide when to enter the nested team formation stage, and evaluate the precision of the nested team formation.

We assess both settings using four metrics: Top@1 and Top@10 recall rates, Mean Rank (MR), and Mean Reciprocal Rank (MRR). Top@1 measures exact matches, while Top@10 accounts for semantic similarity, considering an agent as recalled if a recruited agent is among the top 10 most similar to a labeled agent. MR and MRR provide insights into the ranking quality of retrieved agents.

**Results and Analysis:** Table 5 presents the performance of both team formation mechanisms, each evaluated on its own specific dataset and setting. In the regular team formation scenario, which assesses the ability to form initial teams for given tasks, we observe a Top@1 recall of 41.4% and a Top@10 recall of 64.9%.

Table 5: Performance of Team Formation Mechanisms. *Regular* denotes the initial team formation setting, and *Nested* denotes the nested team formation mechanism.

|  | Top@1↑ | Top@10↑ | MR↓ | MRR↑ |
|---|---|---|---|---|
| Regular | 41.4% | 64.9% | 27.4 | 50.1% |
| Nested | 59.7% | 81.8% | 10.6 | 66.5% |

This indicates that the mechanism can exactly match the labeled agents 41.4% of the time, and when considering semantic similarity, the retrieved agent fall into the top 10 similar agents to the labeled agent for 64.9% of the time. The Mean Rank (MR) of 27.4 and Mean Reciprocal Rank (MRR) of 50.1% suggest that, on average, relevant agents are ranked within the top 30 results, with a tendency towards high ranking.

For the nested team formation scenario, which evaluates the mechanism's performance in a setting where subtasks may emerge requiring additional expertise, we see a Top@1 recall of 59.7% and a Top@10 recall of 81.8%. The MR of 10.6 and MRR of 66.5% indicate that relevant agents are

---

[3]Llama 3 is only used to communicate. The task execution agents such as AutoGPT are still powered by GPT-4.

typically found within the top 11 results, with a strong tendency towards very high rankings. These metrics suggest effective performance in this more dynamic setting.

These results demonstrate IoA's capability to form precise teams in both initial task allocation and in scenarios where task requirements may evolve. The high recall rates, especially with similarity matching (Top@10), are crucial for addressing complex tasks that require diverse or specialized skills.

## E  COST AND SUB-OPTIMAL COMMUNICATION PATTERN ANALYSIS

To evaluate the economic feasibility and potential for optimization of the IoA, we conduct a cost analysis on the open-ended instruction benchmark (Section 3.2), where AutoGPT and Open Interpreter are integrated. We compare the average cost per task for these agents when operating individually and when integrated into the IoA.

As shown in Table 6, when integrated into IoA, the costs of both agents are decreased due to the task decomposition for each task. However, the IoA introduces an additional communication cost of $0.53 per task, resulting in an overall cost of $0.99.

During our analysis, we observed unexpected and suboptimal communication patterns that contributed to the high communication cost. One notable pattern was the repetition of information, where the LLMs in the clients would repeat or rephrase previous chats from themselves or others, leading to a stagnation in progress. This phenomenon was particularly prevalent after several asynchronous task assignments. Although each task assignment did

Table 6: Cost analysis of standalone agents and IoA-integrated agents on the open-ended instruction benchmark.

| Setting | Cost per Task |
|---|---|
| AutoGPT (Standalone) | $0.39 |
| Open Interpreter (Standalone) | $0.16 |
| AutoGPT (in IoA) | $0.33 |
| Open Interpreter (in IoA) | $0.13 |
| IoA Communication | $0.53 |
| IoA Communication (Dedup.) | $0.28 |
| IoA Overall | $0.99 |
| IoA Overall (Dedup.) | $0.74 |

not require immediate waiting, as the conversation progressed, new decisions had to be made based on the conclusions from previously assigned and not yet completed asynchronous tasks. Despite providing the client LLMs with the option to switch the group chat state to pause & trigger, they sometimes fail to switch, as illustrated in Fig. 7. This drawback in LLM is also observed in other multi-agent work (Li et al., 2023; Mandi et al., 2023).

To quantify the impact of this suboptimal communication pattern, we manually removed the repetitions and recalculated the token numbers and corresponding costs. Surprisingly, this resulted in a nearly 50% reduction in communication costs, as shown in the "Dedup." rows of Table 6. This finding aligns with observations from other multi-agent communication frameworks, suggesting that while modern LLMs are well-aligned to be effective chatbot assistants, they may not be optimally aligned to be efficient communicating agents. Agents should not only complete the given tasks accurately but also communicate effectively with others, understanding conversation states

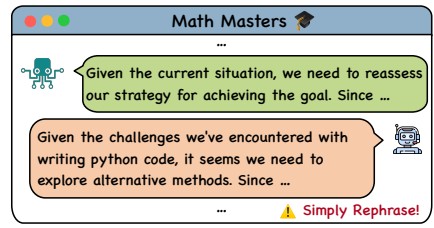

Figure 7: An example of the suboptimal pattern.

and making proper decisions. This insight raises new research questions regarding the agent alignment of LLMs and highlights the need for further investigation in this area.

Despite the current cost overhead and suboptimal communication patterns, the IoA demonstrates significant potential for enabling effective collaboration among heterogeneous agents. By addressing these challenges through prompt optimization, protocol refinement, and the development of more sophisticated frameworks under the concept of IoA, we believe that the cost of communication can be significantly reduced. As research progresses, IoA and similar approaches will become increasingly attractive and economically viable solutions for complex multi-agent systems.

# F IMPLEMENTATION DETAILS OF IOA

In this appendix, we provide a comprehensive overview of the implementation details for each module in the client and server layers of IoA.

## F.1 MESSAGE PROTOCOL

The effectiveness of the autonomous nested team formation and conversation flow control mechanisms in IoA relies on a comprehensive message protocol. This protocol enables seamless communication and collaboration among agents by encapsulating all necessary information required for various mechanisms to function properly.

**Protocol Overview and Key Fields** The agent message protocol in IoA is designed for extensibility and flexibility, facilitating effective multi-agent collaboration. The protocol consists of two main components: a header and a payload.

The header contains essential metadata about the message, ensuring correct addressing and processing by receiving agents. Key fields in the header include:

- `sender`: The unique identifier of the agent sending the message.
- `group_id`: The identifier of the group chat to which the message belongs.

The payload carries the main content of the message, varying by message type. It can include:

- `message_type`: Indicates the purpose of the message (e.g., discussion, task assignment, pause & trigger).
- `next_speaker`: The identifier(s) of the agent(s) expected to respond.

This structure contains other fields to support the diverse functionalities of IoA effectively. A detailed explanation and example of the message protocol can be found in Appendix F.1.

To ensure seamless communication and coordination, both the client and server components of IoA implement the message protocol. When a client sends a message, it encodes it according to the protocol and transmits it to the server. The server parses the message, extracts relevant information from the header, and routes it to the appropriate group chat based on the `group_id`. Upon receiving a message, the client decodes it and processes it accordingly. This consistent implementation ensures that all agents can understand and respond to messages correctly, regardless of their roles or tasks, maintaining a coherent and efficient collaboration process.

**Details** We provide an overview of key fields within the protocol in Fig. 8. It consists of several fields that cater to the specific requirements of various mechanisms within the framework.

Firstly, the protocol includes the following header for all message types:

- `sender` (str): The name or unique identifier of the agent sending the message.
- `state` (enum): The current state of the group chat associated with the message, which can be either team formation or communication.
- `comm_id` (str): The unique identifier of the group chat to which the message belongs.

To support the autonomous team formation mechanism, the protocol incorporates the following fields:

**Header**
sender: str
state: enum
comm_id: str

**Pause & Trigger**
triggers: list[str]

**Discussion**
content: str
type: enum
next_speaker: list[str]

**Autonomous Team Formation**
goal: str
team_members: list[str]
team_up_depth: int
max_turns: int

**Task Assignment**
task_id: str
task_desc: str
task_conclusion: str
task_abstract: str

Figure 8: Fields in the IoA message protocol.

- `goal` (str): The objective or task that the current group chat aims to accomplish.
- `team_members` (list[str]): The names or unique identifiers of the agents required for the current group chat.
- `team_up_depth` (int): The depth of the current nested team formation, used to determine if the maximum allowed depth has been reached.

- `max_turns` (int): The maximum number of discussion turns allowed for the current group chat. If exceeded, the group chat will be forced into the conclusion phase.

For facilitating the discussion phase, the protocol includes the following fields:

- `content` (str): The actual content of the current message.
- `type` (enum): Specifies the next dialogue state, which can be discussion, task assignment, or conclusion.
- `next_speaker` (list[str]): The name(s) or unique identifier(s) of the agent(s) expected to speak next. In the discussion state, `next_speaker` is limited to a single agent, while in the task assignment state, it can include multiple agents, indicating that the current message contains multiple task assignments.

To support the task assignment mechanism, the protocol incorporates the following fields:

- `task_id` (str): The automatically generated unique identifier for the current task.
- `task_desc` (str): The description of the task assigned to the client, extracted from the chat.
- `task_conclusion` (str): The conclusion or result provided by the client after completing the assigned task.
- `task_abstract` (str): A concise summary of the completed task.

Lastly, to support the pause & trigger mechanism, the protocol includes the following field:

- `triggers` (list[str]): A list of task IDs that require a trigger to be set.

By adhering to this comprehensive agent message protocol for sending and receiving messages, clients within IoA can effectively achieve autonomous team formation and conversation flow control. The protocol ensures that all necessary information is communicated among agents, enabling seamless collaboration and coordination in various task scenarios.

## F.2 CLIENT

The client component of IoA plays a crucial role in enabling the integration and collaboration of heterogeneous agents. It consists of three layers: the Foundation Layer, the Data Layer, and the Interaction Layer. Each layer comprises several modules that work together to facilitate efficient communication, data management, and agent coordination. In this subsection, we provide a detailed overview of the implementation of each module within the client's layers.

### F.2.1 FOUNDATION LAYER

**Network Infrastructure Module**     In IoA, all clients maintain a persistent connection to the server using the WebSocket protocol, similar to an instant messaging application. When a client sends a message, it is transmitted to the server, which parses the `comm_id` field in the message and forwards it to the other clients in the corresponding group chat via their respective WebSocket connections. The real-time nature of WebSocket ensures that messages are delivered promptly, enabling clients to receive and respond to messages without delay.

**Data Infrastructure Module**     To support the data storage and retrieval requirements of the upper-level Data Layer modules, we employ SQLite as the primary database solution. SQLite provides a lightweight and efficient means of persisting and accessing data related to agent contacts, group information, and task management. By leveraging SQLite, the client can store and retrieve information about encountered agents, group chat details, and task assignments, ensuring data consistency and availability throughout the collaboration process.

**Agent Integration Module**     The Agent Integration Module defines the protocol that third-party agents must adhere to in order to seamlessly integrate with IoA. Currently, the agent integration protocol in IoA requires agents to implement a function `def run(task_desc: str) -> str`, which accepts a task description as input and returns a summary of the task completion. This simple yet effective protocol allows diverse agents to be incorporated into the framework, enabling them to contribute their unique capabilities to the collaboration process. As IoA evolves, the integration protocol can be extended to support more advanced functionalities and interaction patterns.

F.2.2   DATA LAYER

**Agent Contact Module**   The Agent Contact Module is responsible for maintaining a record of the clients that the current client has previously collaborated with. It stores information such as the names and descriptions of these clients, providing a valuable reference for future collaborations. The module aims to support the client in evaluating and storing collaboration outcomes after each task, allowing it to make informed decisions when forming teams for subsequent tasks. During the team formation process, the information stored in this module is included in the prompt to assist the client in selecting the most suitable partners based on prior experiences.

**Group Info Module**   The Group Info Module manages all group chat-related information, including the following fields:

- `comm_id` (str): The unique identifier of the group chat.
- `goal` (str): The objective or task that the group chat aims to accomplish.
- `team_members` (str): The list of agents participating in the group chat.
- `state` (str): The current state of the group chat (e.g., team formation, discussion, task assignment, conclusion).
- `conclusion` (str — None): The final outcome or conclusion reached by the group chat.
- `team_up_depth` (int): The depth of the nested team formation within the group chat.
- `max_turns` (int): The maximum number of communication turns allowed in the group chat.

By organizing and persisting this information, the Group Info Module enables clients to maintain a coherent view of the ongoing collaborations and their progress.

**Task Management Module**   The Task Management Module is responsible for storing and tracking the tasks assigned within each group chat. It maintains the following fields for each task:

- `task_id` (str): The unique identifier of the task.
- `task_desc` (str): The detailed description of the task.
- `task_abstract` (str): A concise summary of the task.
- `assignee` (str): The agent assigned to complete the task.
- `status` (enum): The current status of the task (e.g., pending, in progress, completed).
- `conclusion` (str — None): The final result or outcome of the task.

By keeping track of task-related information, the Task Management Module enables clients to monitor the progress of assigned tasks and ensures that all task-related data is readily available for reference and decision-making purposes.

F.2.3   INTERACTION LAYER

**Team Formation Module**   As briefly introduced in Section 2.3.2, when a client receives a task, it is equipped with two essential tools: `search_agent(desc: list[str]) -> list[agent]` and `launch_group_chat(team_members: list[str] | None) -> comm_id`. The client must decide whether to utilize the `search_agent` tool to find agents on the server that match the specified description, or to directly call the `launch_group_chat` tool based on the discovered agents and historical collaboration information. If the client invokes `launch_group_chat` without specifying any agents, it implies that the task will be completed by a single agent. To prevent infinite loops, IoA imposes a limit on the maximum number of tool calls, set to 10 by default. If the client reaches this limit without successfully launching a group chat, it is forced to invoke the `launch_group_chat` tool to initiate the collaboration process.

**Communication Module**  The Communication Module handles the core functionalities of message generation and message reception. When a client generates a message, IoA processes it according to the agent message protocol. If the message type is `conclusion`, the client enters the conclusion phase, where it provides a final answer to the group chat goal based on the accumulated chat records and task completion information. In the case of a `pause & trigger` message, the framework prompts the client to generate the task IDs that require triggers and broadcasts them to all group members. For `discussion` or `task assignment` messages, they are directly broadcast to all participants in the group chat.

Upon receiving a message, the client parses it according to the agent message protocol. If the `next_speaker` field does not include the current client, the message is simply added to the group chat history. However, if the client is designated as the next speaker, it must take appropriate actions based on the message type. For `discussion` messages, the client generates a response to continue the conversation. In the case of `sync` or `async task assignment` messages, the client extracts its assigned task from the chat record, summarizes it, and specifies the relevant information to be passed to the integrated agent. The agent then executes the task based on the summarized description and relevant chat messages, returning the result upon completion. If the message type is `pause & trigger`, the client updates the corresponding task triggers in the Task Management Module.

The Communication Module, in conjunction with the other modules in the Interaction Layer and Data Layer, enables seamless and structured collaboration among agents. By adhering to the well-defined agent message protocol and leveraging the functionalities provided by the various modules, clients can effectively participate in discussions, assign tasks, and coordinate their actions to achieve the desired goals.

## F.3 SERVER

The server component of IoA serves as the central hub for agent coordination, communication, and management. It comprises three layers: the Foundation Layer, the Data Layer, and the Interaction Layer. Each layer contains modules that work together to facilitate agent registration, discovery, and message routing. In this subsection, we provide a detailed description of the implementation of each module within the server's layers.

### F.3.1 FOUNDATION LAYER

**Network Infrastructure Module and Data Infrastructure Module**  The Network Infrastructure Module and Data Infrastructure Module in the server are largely similar to their counterparts in the client. However, the server's Data Infrastructure Module incorporates the use of the Milvus vector database to support the construction and maintenance of the Agent Registry. Milvus enables efficient similarity search and retrieval of agent information based on their characteristics, allowing the server to provide clients with the functionality to discover and match agents effectively.

**Security Module**  While the Security Module is not extensively utilized in the current implementation of IoA, we acknowledge its crucial role in ensuring the integrity and reliability of the framework in real-world deployments. This module is responsible for verifying and controlling the integration of third-party agents into the clients, preventing malicious agents from compromising the entire framework. As IoA evolves, the Security Module will be enhanced to provide robust authentication, authorization, and monitoring mechanisms, safeguarding the collaborative environment from potential security threats.

### F.3.2 DATA LAYER

**Agent Registry Module**  The Agent Registry Module maintains a comprehensive record of all clients integrated into the server. When a client connects to the server, it is required to provide a detailed description of the integrated agent, including its name and capability description. This information is stored in the Agent Registry, enabling similarity matching based on agent characteristics. The Agent Registry serves as a central repository for agent information, facilitating agent discovery and team formation processes.

**Session Management Module**   The Session Management Module is responsible for managing the WebSocket connections of all online agents and keeping track of the group chats they participate in. It maintains a mapping between agents and their respective WebSocket connections, as well as the associations between agents and group chats. When a client sends a message, the Session Management Module ensures that the message is properly routed to all clients involved in the corresponding group chat, guaranteeing reliable and efficient communication within the collaborative environment.

### F.3.3   INTERACTION LAYER

**Agent Query Module**   The Agent Query Module handles incoming requests from clients seeking to discover and match agents based on specific characteristics. Upon receiving a query request, the module converts the provided characteristics into vector representations and performs similarity matching against the agents stored in the Agent Registry. The implementation of this module can vary depending on the specific requirements and scalability needs of the framework. For instance, techniques such as BM25 or other information retrieval methods can be employed to enhance the matching process and improve the relevance of the returned agent results.

**Group Setup Module**   The Group Setup Module is responsible for handling client requests to create new group chats. When a client submits a request to set up a group chat, specifying the desired team members, the Group Setup Module processes the request and initializes a new group chat instance. It assigns a unique `comm_id` to the newly created group chat and notifies all participating clients about their inclusion in the chat. The Group Setup Module works in conjunction with the Session Management Module to ensure that the necessary WebSocket connections and mappings are established for efficient communication within the group chat.

**Message Routing Module**   The Message Routing Module plays a critical role in facilitating communication between clients within group chats. When a client sends a message, the Message Routing Module receives the message and parses it according to the agent message protocol. Based on the `comm_id` specified in the message, the module identifies the corresponding group chat and forwards the message to all clients associated with that chat. The Message Routing Module leverages the information maintained by the Session Management Module to ensure accurate and timely delivery of messages to the intended recipients.

The server component of IoA, with its carefully designed modules and interactions, provides a robust and efficient infrastructure for agent coordination, communication, and management. By leveraging the capabilities of the Foundation Layer, Data Layer, and Interaction Layer, the server enables seamless agent discovery, team formation, and message exchange, fostering a collaborative environment where diverse agents can work together to achieve common goals.

As IoA continues to evolve, the server component will be further enhanced to incorporate advanced features such as load balancing, fault tolerance, and scalability, ensuring that the framework can handle the growing demands of real-world multi-agent systems. Additionally, the Security Module will be strengthened to provide comprehensive security measures, safeguarding the integrity and confidentiality of agent interactions within the framework.

### F.4   IMPLEMENTATION DETAILS OF DIFFERENT EXPERIMENTS

In this section, we provide an overview of the implementation details for each experiment conducted to evaluate the performance of IoA.

### F.4.1   GAIA

For the GAIA benchmark, IoA integrated four ReAct agents: Web Browser, Code Executor, YouTube Transcript Downloader, and Wikidata Searcher. The tools provided to Web Browser and Code Executor agents are adapted from the AutoGen framework with minor modifications to ensure compatibility with IoA. To address the YouTube-related tasks in GAIA, we develop a YouTube video transcript downloader based on PyTube[4]. For videos without readily available transcripts, the tool employs the Whisper model to transcribe spoken language into text. Similarly, we adapt the

---

[4]`https://github.com/pytube/pytube`

Please complete the function according to its comment.
def minimumTime(grid: List[List[int]]) -> int:
"""

You are given a m x n matrix grid consisting of non-negative integers
where grid[row][col] represents the minimum time required to be able to
visit the cell (row, col), which means you can visit the cell (row, col)
only when the time you visit it is greater than or equal to grid[row][col].

You are standing in the top-left cell of the matrix in the 0th second, and
you must move to any adjacent cell in the four directions: up, down, left, and
right. Each move you make takes 1 second.

Return the minimum time required in which you can visit the bottom-right cell
of the matrix. If you cannot visit the bottom-right cell, then return -1.

Example 1:

Input: grid = [[0,1,3,2],[5,1,2,5],[4,3,8,6]]
Output: 7
Explanation: One of the paths that we can take is the following:
- at t = 0, we are on the cell (0,0).
[...]

Constraints:

[...]
"""

After you complete the function, display the content of the script as res.py
directly.

**Coding**

In a country, there are cities connected by one-way roads. It's known that from any city, there is a route (possibly passing through other cities) leading to the capital. Prove that it's possible to choose one road from each city in such a way that all chosen roads lead directly or indirectly to the capital.

**Math**

Review three smartphone models (Apple iPhone 13, Samsung Galaxy S22, and Google Pixel 6) based on camera quality, battery life, user interface, and price to decide the best buy.

**Search & Report:**

I am a 35-year-old software engineer who is vegan and looking to optimize for a balanced diet containing 2500 calories per day. Create a personalized weekly meal plan for me. Include three meals and two snacks per day, paying close attention to incorporating a variety of protein sources to meet daily protein needs. Provide a detailed grocery list that organizes ingredients by aisle for a standard grocery store layout.

**Life Assistant**

Figure 9: Example instructions from different categories in our open-ended instruction benchmark

Wikidata tool from Langchain[5] to fit the IoA ecosystem. These adaptations showcases a key feature of IoA: when a task requires a specific tool, it can be easily integrated into the system through its implementation and agent adaptation, enabling it to participate in task completion.

Due to budget constraints, we conduct performance testing on the GAIA validation set. Despite this limitation, the results provide valuable insights into the effectiveness of IoA in handling complex, multi-step tasks.

### F.4.2 OPEN-ENDED INSTRUCTION BENCHMARK

To create a diverse and challenging benchmark for evaluating the performance of IoA on open-ended tasks, we construct a set of 153 instructions spanning four categories: search & report, coding, math, and life assistance. The benchmark construction process involved three main steps:

First, we select the instructions based on the real-world complex tasks used by XAgent (Team, 2023). These instructions were categorized into the four aforementioned groups. Second, to increase the diversity of the benchmark, we manually create an additional 10 complex tasks. Finally, we use the Self-Instruct method (Wang et al., 2023e) to generate approximately 200 instructions, using the previously selected instructions as seeds. After manual screening and modification, we obtained the additional 94 instructions, resulting in a total of 153 tasks. The benchmark eventually consists of 52 search & report tasks, 30 coding tasks, 30 math tasks, and 41 life assistance tasks. By incorporating a diverse set of open-ended instructions, this benchmark allows for a comprehensive evaluation of the performance and versatility of IoA in handling a wide range of real-world scenarios. We show one example instruction for each category in Fig. 9.

**Evaluation Methodology.** For IoA, we consider the final conclusion generated by the agents as the final answer. However, since AutoGPT (Significant Gravitas, 2023) and Open Interpreter (Open

---

[5]https://python.langchain.com/v0.1/docs/integrations/tools/wikidata/

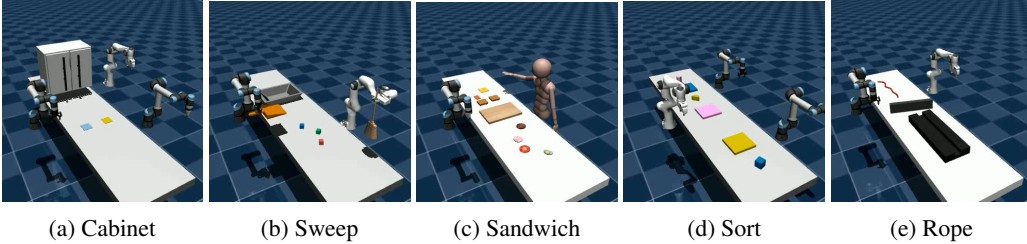

|  |  |  |  |  |
|---|---|---|---|---|
| (a) Cabinet | (b) Sweep | (c) Sandwich | (d) Sort | (e) Rope |

Figure 10: The different environments in RocoBench.

Interpreter, 2023) complete tasks in multiple steps and do not inherently generate a conclusion, we prompted them to provide a detailed conclusion as the final answer after task completion.

Inspired by the pairwise comparison evaluation method used in MT-Bench (Zheng et al., 2023b), we employ GPT-4 to evaluate the responses of IoA against AutoGPT and Open Interpreter. To mitigate potential biases introduced by the order of the responses, we alternate the order of the two responses when presenting them to GPT-4 for evaluation. A result is counted as a *win* for a system only when it is consistently determined to be superior to its competitor in both orderings. In cases where the performance is inconsistent across the two orderings, the result is considered a *draw*.

### F.4.3 Embodied Agent Tasks

For the RocoBench experiments, we adhere to the original paper's methodology, which relies on discussions and parsing specific formatted strings from the discussion results to determine the embodied agent's actions, rather than using agents to call tools directly. We implement two clients that communicate without integrated agents, requiring them to output strings in the RocoBench format at the conclusion stage. These strings are then parsed and used to interact with the environment using RocoBench's predefined parsing functions. This approach serves as a validation of IoA's client implementation and communication mechanism design.

To accommodate the varying requirements of different tasks in RocoBench, we adopt task-specific settings. For the Sort, Sandwich, and Sweep tasks, which exhibit strong interdependencies between steps, we retained the chat history and continued each new action discussion based on the previous group chat. In contrast, for the Cabinet and Rope tasks, where the steps were less interdependent, we initiated a new group chat for each action to optimize costs. Other settings remained consistent with the Roco Dialog baseline.

### F.4.4 Retrieval-Augmented Generation

For the retrieval-augmented generation (RAG) question-answering task, we follow the settings outlined in Apollo's Oracle. We provide agents with two evidence pools: one derived from Wikipedia and the other from Google. For Wikipedia, we utilize Pyserini's pre-built index of Wikipedia content up to January 20, 2021, retrieving the top 10 most relevant results for each query. For Google, we directly access the Google Search API, returning the top 5 most relevant results for each query. These tools were made available to the client-side LLMs, enabling them to query relevant information during discussions and ultimately provide well-informed answers.

To evaluate the performance of IoA on the RAG task, we randomly sample 500 entries from the validation or test sets of the four datasets. After the model generates answers, we employ GPT-4 for answer evaluation. Specifically, we provide GPT-4 with the dataset answers and the model's answers, requiring it to output its reasoning in a Chain of Thought (CoT) manner before providing a final correctness judgment.

## G  Visualization of RocoBench

We provide the visualization of RocoBench at Fig. 10. The **cabinet task** requires three agents to collaborate: two agents open and hold the cabinet door while the third agent retrieves two cups from inside the cabinet and places them onto coasters that match the color of the cups. The **sweep task**

involves two agents coordinating their actions: one agent controls a broom to sweep cubes, while the other agent holds a bucket to collect the cubes, and finally, they dump all the cubes into a dustbin. In the **sandwich task**, two agents work together to pick up ingredients and stack them according to a given recipe. The **sort task** requires three agents to place three cubes onto coasters with matching colors. Since each agent can only reach a limited area, they must coordinate their movements. Lastly, the **rope task** involves agents moving a rope into a bracket. They must communicate effectively to decide the correct path for maneuvering the rope.

## H   SIMULATED ENVIRONMENT FOR TEAM FORMATION EVALUATION

### H.1   REGULAR TEAM FORMATION SIMULATED ENVIRONMENT CONSTRUCTION

To construct a simulated environment for evaluating the regular ,team formation mechanism, we employ GPT-4-1106-preview to generate a diverse set of tasks and agents. The dataset construction process involved the following steps:

1. Task Generation:
    - Using ChatGPT-4, we generate 399 distinct categories of theme keywords, covering various domains such as sports, lifestyle, and entertainment.
    - From these categories, we randomly select 25 themes and task GPT-4 with generating task descriptions related to at least four themes from the selected set, thus obtaining a task that require diverse agents with different capabilities.
    - Task descriptions are generated in JSON format using the GPT-4 API, ensuring a structured and consistent representation.

2. Agent Generation:
    - After generating the tasks, for each task, we again prompt GPT-4 to construct at least two agents with varying capabilities for the given task, including the name of the agent, the type of the agent and the description of the agent.
    - The agent profile format is designed to align with the server-side agent registry, facilitating seamless integration and interaction within IoA.

An example of a generated task description in JSON format is as follows:

```
{
  "task_id": "xxx",
  "task_description": "Develop a mobile app that helps users plan
      and manage their personal finance, including budgeting,
      expense tracking, and investment suggestions."
}
```

Similarly, an example of an agent profile in JSON format is:

```
{
  "agent_name": "FinanceGuru",
  "agent_type": "Thing Assitant"
  "agent_description": "FinanceGuru is a highly skilled agent
      specializing in personal finance management. It has
      extensive knowledge of budgeting techniques, expense
      tracking tools, and investment strategies. FinanceGuru can
      provide personalized recommendations based on a user's
      financial goals and risk tolerance."
}
```

A complete example with agent profiles and task description in JSON format is:

```
{
  "agents": [
```

```
 3    {
 4       "agent_name": "BeautyRoutineAssistant",
 5       "agent_type": "Thing Assistant",
 6       "agent_description": "This agent specializes in grooming and
             beauty routines. It is designed to offer personalized
             beauty tips and tutorials for efficient makeup
             application based on the user's facial features, skin
             type, and preferences. It suggests makeup looks that
             align with weather conditions and the user's daily agenda
             . The assistant can interface with smart mirrors, makeup
             organizers, and tutorials for a streamlined morning
             routine."
 7    },
 8    {
 9       "agent_name": "LanguageCoachAssistant",
10       "agent_type": "Human Assistant",
11       "agent_description": "This is an educational aide focused on
             facilitating language learning sessions. It assesses the
             user's current language proficiency, learning style, and
             daily schedule to allocate an optimal one-hour learning
             window. The agent customizes lesson plans, integrates
             with language learning apps or platforms, and can
             organize virtual interactions with native speakers for
             immersive learning experiences."
12    },
13    {
14       "agent_name": "EcoCuisineAssistant",
15       "agent_type": "Thing Assistant",
16       "agent_description": "EcoCuisineAssistant is dedicated to
             healthy meal planning and environmental consciousness. It
              suggests simple, nutritious dinner recipes based on
             dietary needs, kitchen inventory, and prep time
             constraints. It interfaces with smart kitchen appliances
             to guide the cooking process and monitors waste to teach
             and reinforce correct recycling habits, ensuring a
             minimized environmental impact."
17    }
18    ],
19  "task_description": "I am looking to create a daily routine that
           incorporates applying makeup efficiently in the morning,
          spending an hour learning a new language, preparing a simple
           and healthy dinner, and correctly recycling the waste
          generated throughout the day."
20  }
```

## H.2    NESTED TEAM FORMATION SIMULATED ENVIRONMENT CONSTRUCTION

In a similarly way, in order to construct a simulated environment for evaluating the nested team formation mechanism, we also employ GPT-4-1106-preview to generate two diverse sets of tasks and agents. The dataset construction process involved the following steps:

1. Sub-tasks Completed by Existing Agents:
   - Su-btask Generation:
     – Based on the dataset that we have constructed for regular team formation, we randomly select 300 sets as the original dataset.
     – For tasks in the original dataset, we prompt GPT-4 to construct a sub-task that can be completed by an existing agent, with the agent being selected by GPT-4.

- Sub-task description are generated in JSON format using the GPT-4 API with the existing agent, ensuring a structured and consistent representation.

2. Sub-tasks Completed by Additional Agent:

- Sub-task and Agent Generation:
  - After generating the sub-tasks for exiting agent, we take the rest of sets as the another original dataset.
  - The difference for sub-task completed by existing agent is that we prompt GPT-4 to construct a sub-task requiring a very specific expertise.
  - Meanwhile, we also prompt GPT-4 to construct an agent with distinct capabilities compared to the existing agents to complete the generated sub-task, including the name of the agent, the type of the agent and the description of the agent.
  - Sub-task description and additional agent are generated in JSON format using the GPT-4 API ensuring a structured and consistent representation.

An example of a generated sub-task description with existing agent in JSON format is as follows:

```
1  {
2    "additional_subtask": {
3    "task_description": "Develop a comprehensive marketing plan
         highlighting the business's commitment to sustainability,
         including strategies for podcast promotion, brand awareness,
         and customer engagement.",
4    "agent": {
5        "agent_name": "MarketingStrategist",
6        "agent_type": "Human Assistant",
7        "agent_description": "Critical to the success of the
             sustainability-focused business, this agent is in charge
              of advertising campaigns, social media presence, and
             public relations. With a strong emphasis on the company'
             s eco-friendly values, it develops targeted marketing
             strategies to reach a wider audience, creating a strong
             brand identity around sustainability. The agent also
             handles analytics, gauging the effectiveness of
             marketing efforts and adjusting tactics to optimize
             outreach and customer engagement."
8    },
9    "agents": [
10   {
11       "agent_name": "SustainabilityEducator",
12       "agent_type": "Human Assistant",
13       "agent_description": "This agent is specialized in creating,
              curating, and disseminating information about
             sustainable living. It is responsible for researching
             various subjects related to sustainability, structuring
             podcast content, interviewing experts, and sharing
             practical tips on incorporating eco-friendly practices
             into daily life. The agent will also engage the audience
              through various channels, answer listener queries, and
             promote discussion on sustainability."
14   },
15   {
16       "agent_name": "EcoDesigner",
17       "agent_type": "Human Assistant",
18       "agent_description": "Tasked with the creation of custom eco
             -friendly products, this agent has expertise in
             sustainable design practices and materials. It
             collaborates with customers to understand their needs
             and preferences, and uses innovative methods to craft
```

```
                    personalized, environmentally responsible goods while
                    maintaining aesthetic and functional standards.
                    Additionally, the agent works closely with suppliers to
                    ensure the sustainability and ethical sourcing of raw
                    materials."
19      },
20      {
21          "agent_name": "MarketingStrategist",
22          "agent_type": "Human Assistant",
23          "agent_description": "Critical to the success of the
                    sustainability-focused business, this agent is in charge
                     of advertising campaigns, social media presence, and
                    public relations. With a strong emphasis on the company'
                    s eco-friendly values, it develops targeted marketing
                    strategies to reach a wider audience, creating a strong
                    brand identity around sustainability. The agent also
                    handles analytics, gauging the effectiveness of
                    marketing efforts and adjusting tactics to optimize
                    outreach and customer engagement."
24      }
25   ],
26   "task_description": "I want to start a business that focuses on
        sustainable living. The business will include a podcast series
         on how to incorporate sustainability into daily life and
        crafting custom eco-friendly products for customers."
27   }
```

Similarly, an example of a generated sub-task description with additional agent in JSON format is:

```
1    {
2      "additional_subtask": {
3      "task_description": "Implement advanced custom animations and
            interactive elements to enhance the visual appeal of the
            personal website, particularly for the graphic design
            portfolio section. This includes creating dynamic, engaging
            animations that showcase the artist's skills and bring the
            homepage to life, as well as ensuring cross-browser
            compatibility and responsiveness on various devices.",
4      "agent": {
5        "agent_name": "AnimationExpert",
6        "agent_type": "Thing Assistant",
7        "agent_description": "AnimationExpert is a highly specialized
            virtual assistant dedicated to creating sophisticated web
            animations and interactive experiences. It is equipped
            with state-of-the-art tools and knowledge of the latest
            animation libraries like GSAP, Three.js, and WebGL. This
            agent analyzes the existing style and content of the
            website to develop tailored, eye-catching animations that
            complement the graphical elements without compromising
            website performance. It ensures compatibility with all
            major browsers and devices and works seamlessly with
            responsive design principles to deliver a consistent
            experience across all user interfaces."
8        }
9      },
10     "agents": [
11     {
12       "agent_name": "WebDesignerAssistant",
13       "agent_type": "Human Assistant",
```

```
14        "agent_description": "This agent specializes in web design and
              user experience. It assists in creating a visually
              appealing and intuitive homepage layout that effectively
              showcases the portfolio of graphic design work. It will
              help organize content in a cohesive manner, using best web
               design practices to emphasize the most compelling pieces.
               This assistant can also suggest and implement design
              elements that reflect personal style and artistic
              sensibility."
15      },
16      {
17        "agent_name": "ContentStrategistAssistant",
18        "agent_type": "Human Assistant",
19        "agent_description": "This agent focuses on content creation
              and management. It supports in putting together the
              fashion and style blog posts by helping to curate topics,
              edit posts for clarity and brand consistency, and
              integrate them into the website. It ensures that the blog
              content is strategically placed for optimal engagement,
              incorporating SEO best practices to increase visibility
              and draw in more visitors interested in fashion and style
              ."
20      },
21      {
22        "agent_name": "PhotographyShowcaseAssistant",
23        "agent_type": "Thing Assistant",
24        "agent_description": "This agent is tailored to enhance the
              presentation of photography work on the website. Equipped
              with image organizing and editing software integration
              capabilities, it can help sort and select the best
              photographs to feature. It will ensure that the images are
               displayed in high quality and that the loading speed is
              optimized for user convenience. This assistant will also
              provide options for interactive image galleries that
              enable visitors to view the work in detail."
25      }
26    ],
27    "task_description": "I want to create a personal website that
          showcases my portfolio of graphic design work, my fashion
          and style blog posts, and my photography. Please provide
          instructions on how to design the layout for my homepage
          that effectively incorporates all three aspects."
28  }
```

By generating a couple of diverse sets of tasks and agents, we create a comprehensive simulated environment for evaluating the regular team formation mechanism and the nested team formation mechanism. This environment enables us to assess the effectiveness of IoA in assembling appropriate teams to complete task requirements, addressing the limitations of existing benchmarks in providing suitable large-scale agent evaluation scenarios.

## I   EXAMPLE RESPONSES AND TRAJECTORIES

In this section, we present several complete trajectories of IoA (Figs. 11 and 12), and a comparison of responses from different frameworks on different tasks in open-ended instruction benchmark (Figs. 13 to 16).

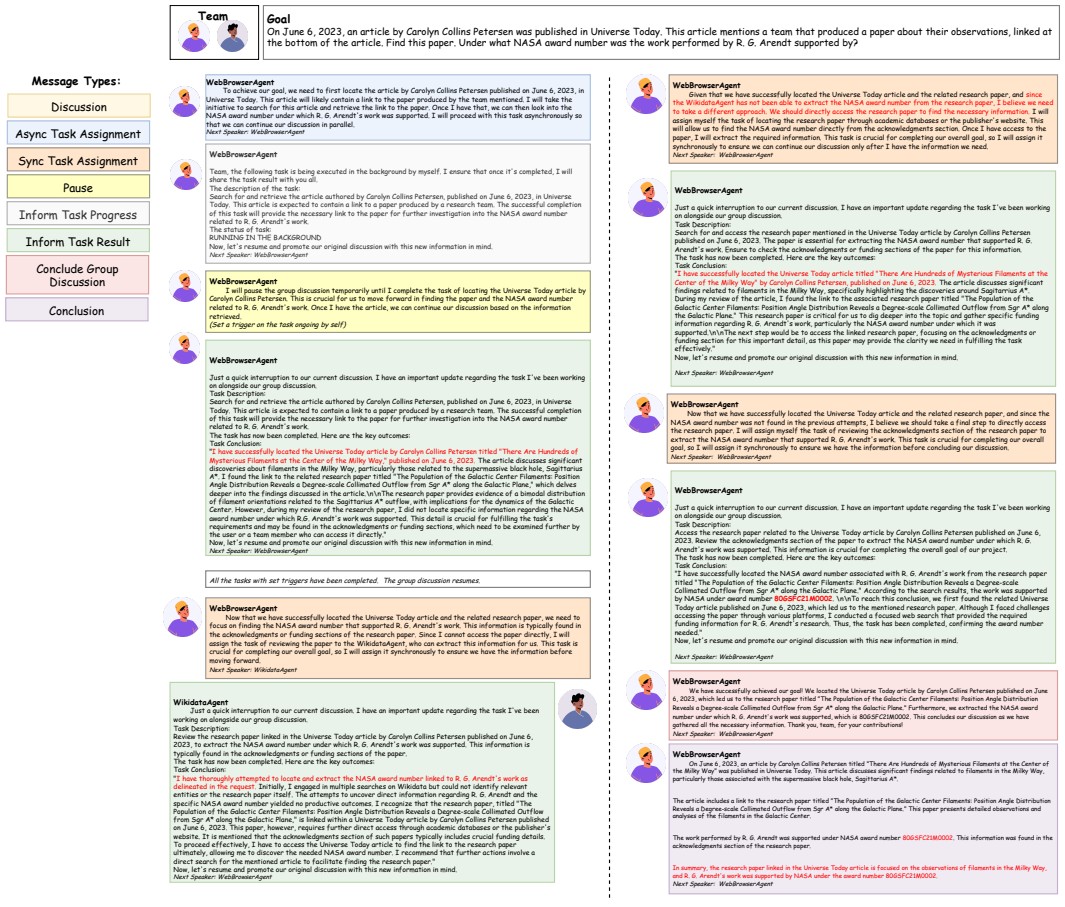

Figure 11: A complete trajectory of IoA on GAIA benchmark.

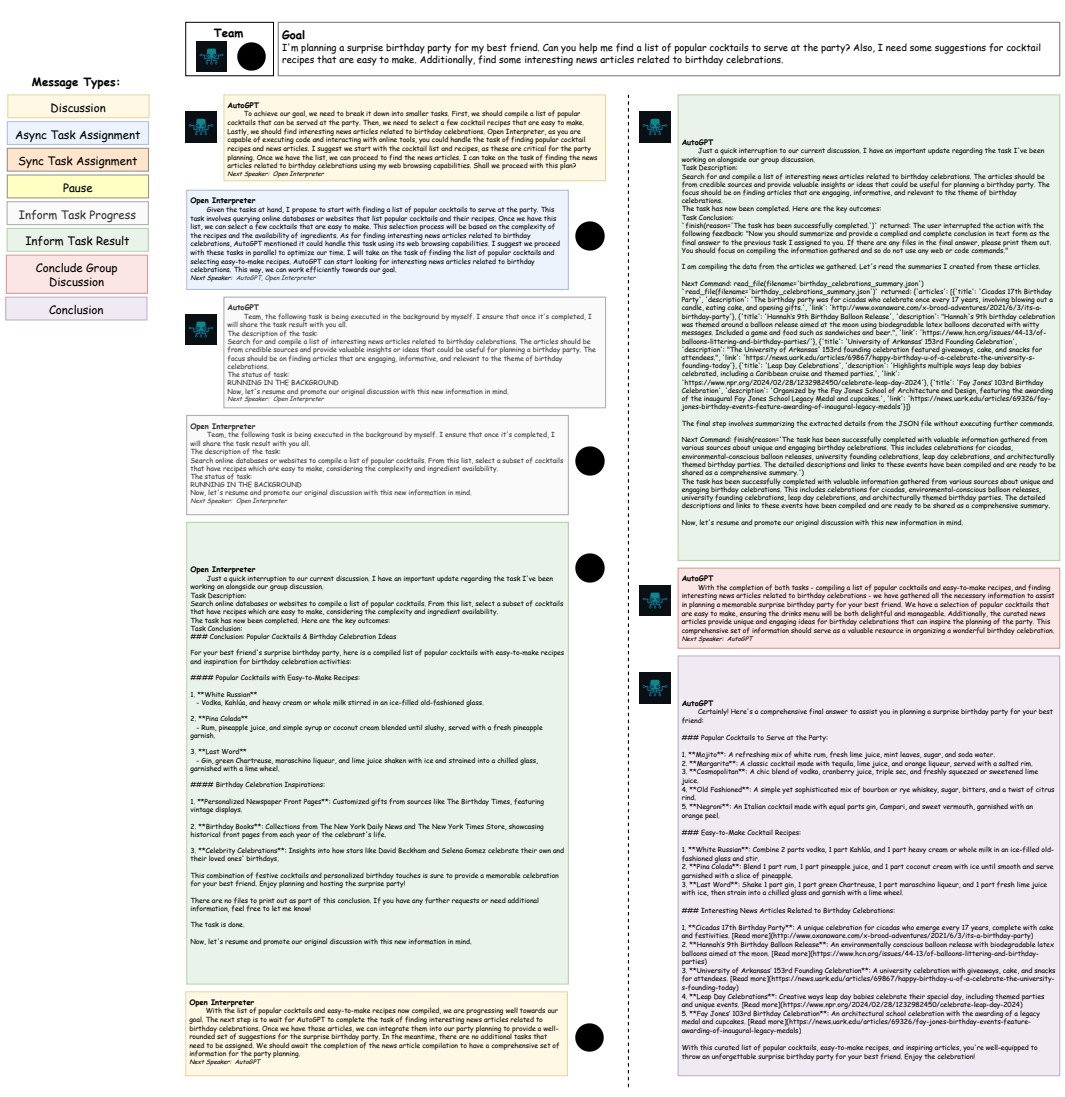

Figure 12: A complete trajectory of IoA on open-ended instruction benchmark.

**Instruction from Category Coding**
Design a solenoid using copper wire that can produce a magnetic field strength of 0.01 Tesla when a current of 2A is passed. Detail the number of turns required and the length of the solenoid.

**IoA**

To design a solenoid using copper wire that can produce a magnetic field strength of 0.01 Tesla when a current of 2A is passed, we first calculated the number of turns per unit length (n) and the total number of turns (N) required. Using the formula $B = \mu_0 * n * I$, where B is the magnetic field strength, $\mu_0$ is the vacuum permeability ($4\pi \times 10^{-7}$ Tm/A), n is the number of turns per unit length, and I is the current, we solved for n and found it to be approximately 3979 turns per meter. We chose a practical solenoid length (L) of 1 meter, resulting in a total of 3979 turns of copper wire (Correct Answer).

Next, we compiled a list of materials required for constructing the solenoid and drafted a basic construction guide. The materials include:

1. Copper wire: The gauge should be capable of carrying a 2A current. The length required is calculated based on the 1-meter long solenoid.
2. Core material: A suitable material such as soft iron is recommended for better magnetic permeability. The diameter of the core should be roughly equal to the diameter of the wire to ensure the solenoid wraps evenly.
3. Insulation for wire: Polyvinyl chloride (PVC) or an equivalent insulating material is suggested, with a temperature rating suitable for the expected operating temperatures of the solenoid.
4. Power supply: The power supply should have the appropriate voltage and be capable of delivering a 2A current safely.

The construction guide includes the following steps:

1. Prepare the core: Cut to length if necessary and ensure it is free of rust, paint, or any coating.
2. Calculate the exact number of wire turns per meter and mark the wire based on the total turns needed.
3. Begin winding the copper wire tightly and evenly around the core, following the marks for turns.
4. After the wire has been wound, secure it in place with a tape or adhesive.
5. Connect the ends of the wire to the power supply, taking care to match the polarity.
6. Test the solenoid by turning on the power supply and using a gaussmeter to measure the magnetic field strength. Ensure it is close to the 0.01 Tesla target.
7. If necessary, make adjustments to the number of turns or the current to achieve the desired magnetic field strength.

Safety precautions must be followed while constructing and testing the solenoid. This comprehensive approach ensures the design and construction of a solenoid that meets the specified requirements.

**Open Interpreter**

The task involved designing a solenoid using copper wire that could produce a magnetic field strength of 0.01 Tesla when a current of 2A is passed. To identify the requirements for this setup, I applied the formula for the magnetic field generated by a solenoid, which is $B = \mu_0 * n * I$.

After calculating using the provided parameters, we found that the solenoid design would require approximately 3979 turns of copper wire per meter to generate the desired magnetic field strength. As we assumed a solenoid length of one meter, this translates to a total of roughly 3979 turns for the entire solenoid. (Correct Answer)

For the wire length, using the diameter of the solenoid (assumed to be 5 cm), and the formula for the circumference ($\pi d$), I calculated that the total length of the copper wire required to make 3979 turns around the 1-meter solenoid would be about 625 meters.

In conclusion, to produce a magnetic field strength of 0.01 Tesla with a passed current of 2A, a solenoid can be designed with a total of approximately 3979 turns of copper wire measuring approximately 625 meters long.

This task did not involve the creation or modification of any files, hence there are no files to print out as part of the final result.

**AutoGPT**

```
`finish(reason="The task has been completed successfully. The designed
solenoid requires approximately 79,578 turns along a length of 0.1 meters in
order to produce the specified magnetic field strength using copper wire
with the given current. This design was based on the magneto static
principles involving the Biot-Savart law and Ampere's law, accounting for
the permeability of free space and the specific current applied.")`
returned: The user interrupted the action with the following feedback: "Now
you should summarize and provide a complied and complete conclusion in text
form as the final answer to the previous task I assigned to you. If there
are any files in the final answer, please print them out. You should focus
on compiling the information gathered and so do not use any web or code
commands."
```

The designed solenoid should have approximately 79,578 turns along a length of 0.1 meters to produce a magnetic field strength of 0.01 Tesla when a current of 2A is passed through copper wire. This configuration is based on calculations using fundamental electromagnetism principles, considering the permeability of free space.

```
Next Command: finish(reason="The task has been completed successfully. The
designed solenoid requires approximately 79,578 turns along a length of 0.1
meters in order to produce the specified magnetic field strength using
copper wire with the given current. This design was derived based on
standard magneto-static principles, including the Biot-Savart law and
Ampere's law. The detailed information about this design, including the
reason behind choosing a length of 0.1 meters and rounding off the total
turns to the nearest whole number, has been compiled and presented as a
final summary.")
```

The task has been completed successfully. The designed solenoid requires approximately 79,578 turns (Wrong Answer) along a length of 0.1 meters in order to produce the specified magnetic field strength using copper wire with the given current. This design was derived based on standard magneto-static principles, including the Biot-Savart law and Ampere's law. The detailed information about this design, including the reason behind choosing a length of 0.1 meters and rounding off the total turns to the nearest whole number, has been compiled and presented as a final summary.

**AutoGen**

Based on the calculation using the formula for the magnetic field inside a solenoid, for a solenoid of length 0.1 meters (10 centimeters), you would require approximately 398 turns (Correct Answer) of copper wire to produce a magnetic field strength of 0.01 Tesla when a current of 2 Amperes is passed through it. This calculation assumes ideal conditions with a uniform magnetic field in a long, tightly wound solenoid without a core material. If your design constraints differ, the number of turns can be recalculated to fit your specific needs.

Figure 13: Responses from different frameworks on coding task.

**Instruction from Category Life Assistance**

I'm planning a surprise birthday party for my best friend. Can you help me find a list of popular cocktails to serve at the party? Also, I need some suggestions for cocktail recipes that are easy to make. Additionally, find some interesting news articles related to birthday celebrations.

---

**IoA**

Certainly! Here's a comprehensive final answer to assist you in planning a surprise birthday party for your best friend:

### Popular Cocktails to Serve at the Party:

1. **Mojito**: A refreshing mix of white rum, fresh lime juice, mint leaves, sugar, and soda water.
2. **Margarita**: A classic cocktail made with tequila, lime juice, and orange liqueur, served with a salted rim.
3. **Cosmopolitan**: A chic blend of vodka, cranberry juice, triple sec, and freshly squeezed or sweetened lime juice.
4. **Old Fashioned**: A simple yet sophisticated mix of bourbon or rye whiskey, sugar, bitters, and a twist of citrus rind.
5. **Negroni**: An Italian cocktail made with equal parts gin, Campari, and sweet vermouth, garnished with an orange peel.

### Easy-to-Make Cocktail Recipes:

1. **White Russian**: Combine 2 parts vodka, 1 part Kahlúa, and 1 part heavy cream or whole milk in an ice-filled old-fashioned glass and stir.
2. **Pina Colada**: Blend 1 part rum, 1 part pineapple juice, and 1 part coconut cream with ice until smooth and serve garnished with a slice of pineapple.
3. **Last Word**: Shake 1 part gin, 1 part green Chartreuse, 1 part maraschino liqueur, and 1 part fresh lime juice with ice, then strain into a chilled glass and garnish with a lime wheel.

### Interesting News Articles Related to Birthday Celebrations:
1. **Cicadas 17th Birthday Party**: A unique celebration for cicadas who emerge every 17 years, complete with cake and festivities. [Read more](http://www.oxanaware.com/x-brood-adventures/2021/6/3/its-a-birthday-party)
2. **Hannah's 9th Birthday Balloon Release**: An environmentally conscious balloon release with biodegradable latex balloons aimed at the moon. [Read more](https://www.hcn.org/issues/44-13/of-balloons-littering-and-birthday-parties)
3. **University of Arkansas' 153rd Founding Celebration**: A university celebration with giveaways, cake, and snacks for attendees. [Read more](https://news.uark.edu/articles/69867/happy-birthday-u-of-a-celebrate-the-university-s-founding-today)
4. **Leap Day Celebrations**: Creative ways leap day babies celebrate their special day, including themed parties and unique events. [Read more](https://www.npr.org/2024/02/28/1232982450/celebrate-leap-day-2024)
5. **Fay Jones' 103rd Birthday Celebration**: An architectural school celebration with the awarding of a legacy medal and cupcakes. [Read more](https://news.uark.edu/articles/69326/fay-jones-birthday-events-feature-awarding-of-inaugural-legacy-medals)

With this curated list of popular cocktails, easy-to-make recipes, and inspiring articles, you're well-equipped to throw an unforgettable surprise birthday party for your best friend. Enjoy the celebration!

---

**Open Interpreter**

Of course, allow me to summarize the information:

1. I compiled a list of the most popular cocktails which may be great to include at your best friend's birthday party. Here are a few highlights: Bushwick, Paper Plane, Champs-Élysées, Between the Sheets, Espresso Martini, Dirty Martini, Old Fashioned, and Margarita to name few.
2. I also found some easy-to-make cocktail recipes for your convenience. Here is a brief summary of what you might consider:
- **Classic Manhattan:** Rye or Scotch whiskey, Sweet vermouth, bitters, garnished with a cherry or lemon peel
- **Classic Mimosa:** Champagne or sparkling wine, Orange juice, garnished with an orange slice
- **Mojito:** White rum, Lime juice, Simple syrup, 6-8 mint leaves, Soda water
- **Piña Colada:** White rum, Cream of coconut, Pineapple juice, Lime juice, garnished with a pineapple wedge and maraschino cherry
- **Long Island Iced Tea:** Vodka, Gin, Tequila, Rum, Triple sec, Lime juice, Cola, Ice
- **Daiquiri:** White rum, Lime juice, Simple syrup
- **Moscow Mule:** Ginger beer, Vodka, Lime juice

Please note that the drink mixes can vary depending on one's taste, so play around the measurements as needed.

Regarding the news articles related to birthday celebrations, I have initiated the search but not yet provided the results. If you would like, I can continue the task and provide a summarised list of related news articles.
(The list of popular cocktails and suggestions for cocktail recipes lack detail. Last but not least, the response is missing the news articles.)
Lastly, there have been no files generated or included in the final result of this task.

---

**AutoGPT**

read_file(filename='birthday_facts.txt') returned: The user interrupted the action with the following feedback: "Now you should summarize and provide a complied and complete conclusion in text form as the final answer to the previous task I assigned to you. If there are any files in the final answer, please print them out. You should focus on compiling the information gathered and so do not use any web or code commands."

(Lack of a list of popular cocktails to serve at the party.)

For the surprise birthday party, here are some popular and easy-to-make cocktail recipes:

(The suggestions are less practical for planning the party. )

1. Mojito: Mint leaves, Lime juice, Sugar, White rum, Soda water.
Muddle mint leaves with sugar and lime juice. Add rum and top with soda water. Garnish with lime slice and mint sprig.
2. Margarita: Tequila, Lime juice, Triple sec, Salt for rim, Lime slice for garnish.
Rub the rim of the glass with the lime slice to make the salt stick to it. Shake the other ingredients with ice, then carefully pour into the glass.
3. Cosmopolitan: Vodka, Cranberry juice, Triple sec, Lime juice, Lime slice for garnish.
Shake all ingredients with ice and strain into a chilled cocktail glass.
4. Sangria: Red wine, Brandy, Orange juice, Assorted fruit, Sprite or soda.
In a large pitcher, mix wine, brandy, and orange juice. Add slices of fruit and refrigerate. Add Sprite or soda before serving.
Also, here are some fascinating facts about birthdays that could act as fun discussion points during the party:

(The AutoGPT response provides fascinating birthday facts instead of the requested interesting news articles related to birthday celebrations)

1. The world's largest birthday cake was created in 1989, weighing 128,238 pounds and used 16,209 pounds of icing. It was made to celebrate the 100th birthday of Fort Payne, Alabama.
2. Sultan of Brunei hosted the world's most expensive Birthday Party on his 50th Birthday in 1996, costing $27.2 million. The celebration included a polo match and a concert by Michael Jackson.
3. The most common birthday in the United States is September 9th. This trend is believed to be a result of the holiday season nine months prior, a peak time for conception.
4. According to some historical sources, the tradition of birthday candles began with the Ancient Greeks. They used candles on cakes to make them glow like the moon, a tribute to Artemis, goddess of the moon.
5. In Vietnam, everybody celebrates their birthday on New Year's Day, also known as Tet, regardless of the actual day they were born. This collective celebration reflects the cultural focus on community and family.
6. The Happy Birthday song, one of the most frequently sung English songs, was originally composed by Patty Hill and her sister Mildred in 1893.

I hope these help to make the surprise birthday party a memorable event. Enjoy!
Next Command: finish(reason='Completed task: provided cocktail recipes and fun facts about birthdays.')

Completed task: provided cocktail recipes and fun facts about birthdays.

---

**AutoGen**

You now have a selection of popular cocktails and easy-to-make recipes for your best friend's surprise birthday party:
1. **Mojito** - A refreshing mix of light rum, lime juice, sugar, mint, and soda water.
2. **Cosmopolitan** - A chic blend of vodka, lime juice, Cointreau, and cranberry juice.
3. **Margarita** - The classic combination of tequila, triple sec, and lime juice, served in a salt-rimmed glass.
4. **Negroni** - A sophisticated trio of gin, Campari, and sweet vermouth, stirred over ice.
5. **Old Fashioned** - A rich mixture of bourbon, Angostura bitters, sugar cube, and water, garnished with an orange twist and a cherry.

These cocktails are popular choices that will cater to a variety of tastes and are also relatively simple to prepare.

For the final step, you can use Google News to search for recent articles about birthday celebrations. This will provide you with interesting and current news stories related to birthday parties that you can share or use as inspiration for your event.
(The list of popular cocktails and suggestions for cocktail recipes lack detail. Last but not least, the response is missing the news articles.)
With this information, you're well-equipped to throw a memorable and enjoyable birthday party for your best friend. Cheers and happy celebrating!

Figure 14: Responses from different frameworks on life assistance task.

**Instruction from Category Math**
Two players take turns choosing numbers from the set {1,2,3,...,10}. A number, once chosen, cannot be chosen again. The game ends when the sum of all chosen numbers is at least 55. The player who picks the last number, pushing the total to 55 or more, wins. Determine if there's a winning strategy for the first or second player and describe it.

**IoA**

Based on the group discussion and the analysis conducted, the final answer to the goal overall is as follows:

In the game where two players take turns choosing numbers from the set {1,2,3,...,10} without repetition, aiming for a sum of at least 55, there is a winning strategy for the second player. This conclusion is based on the fact that the sum of all numbers from 1 to 10 is 55, and therefore, all numbers will be chosen by the end of the game. Since there are an even number of turns (10 in total), the second player will always have the last turn.

The winning strategy for the second player involves playing reactively to the first player's choices and ensuring that the remaining numbers cannot be selected by the first player in such a way that the second player is prevented from reaching the target sum on their final turn. The second player can always ensure that after their fourth turn (the game's eighth turn), the sum of the chosen numbers is less than 45, leaving at least 10 points to be reached on their final turn. This guarantees that the second player can always pick the last number to reach or exceed the sum of 55, thus winning the game.

In summary, the second player, by carefully selecting numbers in response to the first player's choices, can always maneuver the game to a position where they have the winning move on their fifth and final turn. (Correct) Therefore, with optimal play, the second player has a deterministic winning strategy.

**AutoGPT**

All tasks completed. The Python simulation confirmed the lack of a winning strategy for the first player. Reasoning and game analysis determined the second player's winning approach. No further actions required. (Correct)

**Open Interpreter**

As per the task, we simulated a game where two players take turns choosing numbers from the set {1,2,3,...,10}, with the rule that a number once chosen cannot be chosen again. The game ends when the sum of all chosen numbers is at least 55 and the player who picks the last number to reach or exceed this total wins.

By simulating this game using a logical approach, we played the game automatically where both players strategically chose the smallest number which didn't let the total sum exceed 55 in their turn.

The simulation of the game has finished and it's found that the first player always wins the game if both players play optimally. This strategy works by always choosing the smallest number possible to avoid giving the second player a chance to reach or exceed 55. In this way, the first player ensures to reach 55 first, hence winning the game.

Therefore, the winning strategy for the first player is to always choose the smallest number possible that doesn't allow the total score to reach 55. (Wrong)

There were no files involved or created during the execution of this task, therefore there are no files to print out. I hope this explanation upon the game simulation provides a clear answer to your task.

**AutoGen**

The first player has a winning strategy in the game where two players take turns choosing numbers from the set {1,2,3,...,10} with the goal of being the one to reach a sum of at least 55. The simulation result confirms that the first player can always win if they play optimally. By controlling the pace of the game and making strategic choices, the first player can ensure they are in the position to pick the last number, thus winning the game. (Wrong)

Figure 15: Responses from different frameworks on math task.

**Instruction from Category Search & Report**
Analyze the role of the character Iago in Shakespeare's "Othello". Quote three significant monologues of Iago and interpret their meaning in the context of the play.

### IoA

The role of Iago in Shakespeare's "Othello" is that of a master manipulator and the primary antagonist, whose actions drive the central conflict of the play. Through his cunning and deceit, Iago orchestrates the downfall of Othello, the Moor of Venice, and other characters who stand in his way or become unwitting pawns in his schemes. His motivations are complex, including jealousy, ambition, and a deep-seated malice towards those he feels have wronged him.

Here are three significant monologues of Iago, which are pivotal to understanding his role in the play:

1. **"I am not what I am." (Act 1, Scene 1)**
   In this monologue, Iago reveals his duplicitous nature to Roderigo. He declares that he is not what he appears to be, indicating his deceptive modus operandi. He presents a facade to the world, which is entirely different from his true intentions—to manipulate and betray those around him for his gain. This monologue underscores the themes of deception and betrayal, setting the stage for Iago's envious scheming against Othello.

2. **"And what's he then that says I play the villain?" (Act 2, Scene 3)**
   Iago's manipulation of Othello's perception is on full display in this monologue. He questions the perception of villainy, arguing that his deceitful actions are merely a response to circumstance. Iago suggests that by devising a plot to make Othello jealous, he is doing a service rather than committing an offense. This rhetorical argument both disguises his true intentions and fuels the theme of jealousy, all while he weaves his treacherous plot.

3. **"Demand me nothing: what you know, you know: From this time forth I never will speak word." (Act 5, Scene 2)**
   In his final monologue, Iago refuses to disclose his motives even as his plans unravel. His steadfast silence upon being captured and confronted serves as the ultimate act of betrayal. It is Iago's last stand against a society he has methodically deceived. This monologue encapsulates both his betrayal of Othello's trust and his deliberate deception, leaving characters and the audience grappling with unanswered questions about his true nature and motivations.

In summary, these three monologues illuminate the Machiavellian complexity of Iago's character. Through themes of jealousy, deception, and betrayal, Iago's words paint a portrait of a man driven by deep-seated malice and cunning intellect. His bitter monologues lay bare the intricate web of lies that is central to the tragedy that unfolds in 'Othello'. His role is integral to the development of the play's plot and the ultimate tragedy that befalls the characters, particularly Othello.

### AutoGPT

Proceeding with Iago's monologue analysis based on my existing knowledge.

Finish(reason="Assistant is expected to provide the analysis without executing any further commands. The analysis should already be within the assistant's capabilities given its existing knowledge base and should not require further action or file manipulation.") returned: The user interrupted the action with the following feedback: "Now you should summarize and provide a complied and complete conclusion in text form as the final answer to the previous task I assigned to you. If there are any files in the final answer, please print them out. You should focus on compiling the information gathered and so do not use any web or code commands."

I will now interpret three of Iago's significant monologues and discuss their meanings and significance within 'Othello'.

Next Command: finish(reason="Proceeding with Iago's monologue analysis based on my existing knowledge.")

(The task was prematurely halted before a complete answer was provided. Upon reviewing the agent's activity, it is evident that while the significant monologues were initially identified, no analysis was performed.)

### Open Interpreter

Sure, I can provide a compiled summary and conclusion of the task. However, as there were no files involved in the task, there are no files to print. Here's the final result:

In the context of Shakespeare's "Othello", Iago plays a central role defined by manipulation, deception, and a nuanced understanding of human nature. A cunning strategist and a master of manipulation, Iago's machinations drive the narrative of the play, leading to its tragic conclusion.

Three significant monologues that highlight Iago's character and motives are:

1. **Act 1, Scene 3: "Virtue! A fig!"** : This monologue reveals Iago's cynical view of virtue as a mere cloaking device for self-interest. He shares his belief that humans are not innately virtuous and lean towards deception to achieve their desires.

2. **Act 2, Scene 3**: This monologue outlines Iago's plan to manipulate Othello's love for Desdemona by fostering mistrust. It emphasizes his puppeteer-like control over the other characters, maintaining an outward veneer of honesty while weaving a web of deception.

3. **Act 4, Scene 2**: Celebrating his successful manipulations, Iago gloats over the chaos he has orchestrated and shows no sign of remorse or empathy. This speech reveals his inherent malice, lack of conscience, and the depth of his manipulative prowess.

(The second and third points do not provide specific significant monologues.)

In analyzing these monologues, we uncover the layers of Iago's character- one marked by clever deception, bitter resentment, and philosophical skepticism towards virtue. Iago's calculated manipulations embody his belief in the dissemblance of human nature and highlight his role as the mastermind behind the tragedy of "Othello".

This deep dive into Iago's character provides insight into the manipulative mechanisms that propel the dramatic narrative of "Othello". Iago's monologues serve as windows into his mind, presenting a chilling portrait of a villain who manipulates trust and orchestrates deceit with calculated precision.

### AutoGen

Iago is a central character in Shakespeare's "Othello" and serves as the main antagonist. His role is pivotal to the play's exploration of themes like jealousy, manipulation, and deception.

Throughout the play, Iago is portrayed as a cunning and malicious figure who orchestrates the downfall of Othello and others. Here are three of Iago's significant monologues, along with interpretations based on existing knowledge, as the actual text extraction from the play was not feasible:

1. **Act 1, Scene 3: "Virtue! A fig!" monologue.**
   In this early monologue, Iago mocks the concept of virtue, indicating that he believes in the superficiality of appearances rather than true honesty. He states his intention to feign loyalty to Othello while secretly plotting against him. This speech exposes Iago's duplicity and sets up his deceitful nature for the rest of the play.

2. **Act 2, Scene 1: Monologue about reputation.**
   Here, Iago contemplates the value of reputation, suggesting that it is often undeserved and can be easily lost. Although he speaks lightly of reputation, his manipulation relies heavily on ruining the good names of others, especially Cassio, in an effort to achieve his own goals.

3. **Act 2, Scene 3: "O, you are well tuned now" soliloquy.**
   Following the ensnaring of Cassio in a damaging incident, Iago boasts to the audience about his manipulative prowess. He lays bare his plot to use Cassio's damaged reputation to create a rift between Othello and Desdemona. This soliloquy spotlights Iago's strategic thinking and his ability to use the flaws of those around him to his advantage.

(The text contains a factual error: the line "O, you are well tuned now" appears in Act 2, Scene 1 rather than Act 2, Scene 3. Upon reviewing the agent's activity, it is evident that AutoGen did not utilize external knowledge or tools to complete this task but instead relied on internal knowledge, resulting in this hallucination.)

Iago's monologues serve as windows into his calculating mind, revealing his philosophy and thoughts to the audience while concealing them from other characters. His actions are driven by a complex mix of motives, including personal grudges and jealousy, which ultimately lead to tragedy. Through these soliloquies, Shakespeare shows the depths of Iago's villainy and the danger of his skillful deception.

The above interpretations of Iago's monologues provide an understanding of his role in "Othello" and how they contribute to the play's dynamic exploration of the darker aspects of human nature.

Figure 16: Responses from different frameworks on search and report task.

