# OpenReview forum: "Internet of Agents: Weaving a Web of Heterogeneous Agents for Collaborative Intelligence"
_ICLR.cc/2025/Conference — ICLR 2025 Spotlight_

### Official Review · Reviewer_LshR · 2024-11-03

**Soundness:** 3
**Presentation:** 4
**Contribution:** 4
**Rating:** 8
**Confidence:** 4

**Summary:**

The authors propose the Internet of Agents (IoA), a novel framework for enabling collaboration between heterogeneous AI agents distributed across different devices. Drawing inspiration from the Internet's architecture, IoA introduces key mechanisms including agent registration/discovery, nested team formation, conversation flow control using finite state machines, and flexible task assignment. The framework addresses limitations of existing multi-agent systems like ecosystem isolation, single-device constraints, and rigid communication pipelines.

**Strengths:**

- The authors present a well-motivated and timely contribution given the rapid advancement of autonomous agents and the need for frameworks to orchestrate their collaboration
- The layered architecture design is elegant and comprehensive, with clear separation between interaction, data, and foundation layers for both server and client components
- The experimental evaluation is thorough, testing the framework across diverse scenarios including general assistance, embodied AI tasks, and retrieval-augmented generation
- The nested team formation mechanism is theoretically sound, with formal analysis showing reduced communication complexity compared to fully connected structures
- The finite state machine approach to conversation control is well-grounded in Speech Act Theory and provides a principled way to manage agent interactions

**Weaknesses:**

- While the authors demonstrate strong empirical results, the theoretical analysis of the framework's properties (e.g., convergence guarantees, scalability limits) could be strengthened
- The evaluation focuses primarily on task performance metrics. Additional analysis of system properties like latency, resource usage, and communication overhead would be valuable (I do see some of this in Appendix E)
- The authors could expand on potential failure modes and mitigation strategies, particularly for scenarios with unreliable network connections or agent failures
- The security implications of allowing distributed agents to collaborate deserve deeper treatment, including potential vulnerabilities and safeguards. I appreciate that the authors included the Security Module in the schematic, but think having a bit more discussion of it (in the appendix) could be beneficial

**Questions:**

- Really cool that you drew inspiration from Speech Act Theory for this work. Have you considered what Rational Speech Act has to offer in this case? Would be neat to see it leveraged to deal with things like recursive reasoning (theory of mind), own uncertainty and uncertainty of other agents, etc.
- How does the system handle potential deadlocks in the conversation state machine? Are there specific examples of potential deadlock scenarios you've considered? Can you describe any mechanisms or safeguards implemented to prevent or resolve such deadlocks in the conversation flow?
- What mechanisms ensure fair resource allocation when multiple agent teams are operating simultaneously?
- How does the framework maintain consistency when agents have conflicting knowledge or goals?
- Could the authors elaborate on the scalability limits of the nested team formation approach?
- What strategies are employed to prevent malicious agents from disrupting team collaboration? Is that just offloaded to the Security Module?

---

> ### Author Response · Authors · 2024-11-20
>
> Thank you for your thoughtful and detailed review, and for recognizing the motivation, design, and contributions of our IoA framework. Below, we address your questions and concerns in detail.
>
> ---
>
> **1. Theoretical Analysis of Properties (Convergence, Scalability)**
>
> Thank you for raising this important point. Still we would like to say that LLM-based agent systems represent a highly empirical research area, and sometimes the primary goal is to develop systems that work effectively in practice, as the field is still in its infancy. While theoretical guarantees such as convergence are desirable, they are inherently challenging to formulate for LLM-based systems. The use of language as the primary medium introduces complexities that are indifferentiable, making it difficult to apply theoretical frameworks like those used in traditional RL research.
>
> In multi-agent systems, these challenges are further compounded by the dynamics of inter-agent communication and collaboration, where properties like convergence are particularly hard to predict or measure. Despite these hurdles, we have made an effort to justify IoA's scalability from the perspective of communication topology. Specifically, the sparse communication structure introduced by the nested team formation mechanism significantly reduces the number of communication edges compared to fully connected architectures. This sparsity is a promising direction for improving the scalability of multi-agent systems, and our empirical results already demonstrate its potential.
>
> We fully acknowledge the importance of further theoretical analysis and recognize this as a shared challenge within the LLM-based multi-agent research community. We will continue exploring ways to incorporate theoretical insights into future iterations of IoA. Thank you for your thoughtful advice, which has helped us reflect on these critical aspects of our framework.
>
> ---
>
> **2. Analysis of Latency, Resource Usage, and Communication Overhead**
>
> Thank you for pointing this out, and we appreciate your careful reading. As noted, Appendix E includes preliminary analysis of costs. As the communication overhead is largely associated with the number of generated tokens and input tokens, the cost can also partially serve as a proxy of communication overhead. On average, IoA is approximately 2 to 3 times slower than standalone agents due to the added complexity of agent collaboration. Still, as mentioned in our response to reviewer 78P6’s first question, we have a large room for optimization. In addition, directly quantifying network latency in distributed settings remains challenging due to varying device configurations and network conditions.
>
> As a prototype implementation of the broader IoA concept, our current focus is on demonstrating performance effectiveness and validating the benefits of connecting heterogeneous, distributed agents. We recognize the importance of analyzing additional metrics, such as latency and real-world operating performance. We will address these aspects in future work with more advanced and comprehensive designs.
>
> ---
>
> **3. Potential Failure Modes, Mitigation Strategies, Security, Deadlock, and Resource Allocation**
>
> Thank you for suggesting a more detailed discussion of failure modes and related aspects. In scenarios with unreliable networks or agent failures, IoA employs the following strategies:
>
> - **Retry Mechanisms**: Agents automatically retry communication with the server or tools when initial attempts fail.
> - **Autonomous Error Handling**: Our experiments demonstrate that agents can handle task failures autonomously, even without explicit instructions. An example of this process is illustrated in the anonymous link https://anonymous.4open.science/r/ioa-rebuttal-606B/complete_trajectory_GAIA.pdf, where a WebBrowserAgent collaborates with a WikidataAgent. When the WikidataAgent fails due to tool limitations, the agents decide to let the WebBrowserAgent attempt the task, ultimately succeeding.
>
> Regarding security and deadlock, we acknowledge that these aspects are currently addressed only at a basic level. For example, simple timeout-based retries are employed to handle potential deadlocks. Additionally, IoA does not impose explicit assumptions on the resources available to each agent. While these are important considerations, our current focus has been on demonstrating task performance effectiveness as a foundational exploration of the Internet of Agents concept.
>
> We agree that IoA's robustness can be significantly enhanced with more advanced error recovery, fault tolerance, and security mechanisms. These, along with strategies for addressing deadlocks, resource allocation, and behavior monitoring, are essential for the “ultimate” version of IoA and will be explored in future iterations. To address your feedback, we will add a section in the appendix outlining these considerations and potential enhancements to improve the system's overall robustness and reliability.

---

> > ### Author Response · Authors · 2024-11-20
> >
> > **4. Leveraging Rational Speech Act Theory**
> >
> > We are glad you found our use of Speech Act Theory interesting! Incorporating RSA Theory to model recursive reasoning and uncertainty is an exciting direction. For example, RSA might enhance task decomposition and coordination by allowing agents to infer others’ intentions or adjust communication strategies dynamically. While this is beyond the current scope of IoA, we see significant potential in combining RSA with IoA for more sophisticated reasoning and team dynamics. We appreciate this suggestion and will explore it in future work.
> >
> > ---
> >
> > **5. Conflicting Knowledge or Goals Among Agents**
> >
> > While IoA does not explicitly address conflict resolution between agents with conflicting knowledge or goals, we observe that the flexibility of the FSM framework, combined with the advanced reasoning capabilities of SoTA LLMs, often enables agents to resolve conflicts autonomously. During the discussion state, agents can deliberate, provide evidence to support their positions, and collaboratively adjust their goals or knowledge to reach a consensus.
> >
> > We acknowledge that explicitly formalizing conflict resolution mechanisms could further enhance IoA’s robustness and reliability, and we see this as a promising direction for future work. Nonetheless, the observed conflict resolution capabilities of agents in IoA already demonstrate its potential to handle such situations effectively in practice.
> >
> > ---
> >
> > **6. Scalability Limits of Nested Team Formation**
> >
> > The nested team formation mechanism leverages a sparse communication topology, allowing IoA to scale effectively with larger numbers of agents. By forming small, localized groups of three to five agents within each newly created team, communication overhead is significantly reduced, as most interactions remain confined to these manageable subgroups. Recently, some concurrent work has also shown benefits of such sparse communication topology [1].
> >
> > However, a key scalability challenge lies in teammate selection. As the number of agents in the system grows, identifying the most suitable collaborators becomes increasingly complex. Agents must efficiently search for others with the desired capabilities, and the system must ensure accurate matching. This complexity can lead to less optimal team formation as the pool of agents expands. To address this, future work will focus on improving capability matching algorithms and developing more robust team optimization strategies to maintain scalability and effectiveness.
> >
> > ---
> >
> > We hope these responses address your concerns and provide clarity on IoA’s mechanisms and safeguards. We sincerely thank you for appreciating our efforts as an initial exploration of this ambitious concept and for recognizing the diverse empirical results we presented. We acknowledge that IoA represents a broad and challenging direction, requiring significant future research into areas such as robustness, security, and resource allocation, many of which cannot be fully addressed within a single paper.
> >
> > Nonetheless, we believe our work demonstrates the potential of this concept and could serve as a foundation for a new research direction in MAS. We are grateful for your recognition of the value of our contributions and for your thoughtful feedback, which will greatly guide our future efforts.
> >
> > [1] Li, Yunxuan, et al. "Improving Multi-Agent Debate with Sparse Communication Topology." *arXiv preprint arXiv:2406.11776* (2024).

---

> ### Comment · Reviewer_LshR · 2024-12-03
>
> Thank you for engaging with my review! I'm very happy to maintain my score and hope to see this paper at the conference!

---

### Official Review · Reviewer_78P6 · 2024-11-03

**Soundness:** 3
**Presentation:** 4
**Contribution:** 3
**Rating:** 8
**Confidence:** 3

**Summary:**

This paper introduces the Internet of Agents (IoA), a framework for enabling distributed collaboration among heterogeneous LLM-based agents. The key innovations include:
(1) a protocol for distributed agent collaboration: a multi-layered architecture that enables agents to operate across different devices and locations, where each agent maintains its own "WebSocket" connection to a central server that handles message routing and coordination.

(2) an instant-messaging-like architecture for agent discovery and teaming: agents can search for collaborators based on capability descriptions and form nested teams, implemented through a central registry that maintains agent capabilities and allows semantic matching for team formation

(3) dynamic mechanisms for conversation flow control: a finite state machine approach that manages conversation states (discussion, task assignment etc.,), where LLMs autonomously decide state transitions and next speakers to maintain structured dialogue

(4) a protocol for integrating third-party agents: a standardized interface that requires agents to implement a simple `run()` function accepting task descriptions and returning results, allowing agents built under different protocols (AutoGPT or OpenInterpreter) to be wrapped and integrated into the framework.

IoA (GPT3.5) is evaluated across GAIA benchmark, open-ended instructions, embodied AI tasks, and retrieval-augmented generation, and IoA consistently outperformed other baselines on GAIA, open-ended instruction benchmark and matches or exceeds GPT4 performance on IoA.

**Strengths:**

1. The architecture is novel and the design is well-motivated which address three key issues in typical multi-agent systems (MAS).
2. The technical foundation is clear as there is rigorous formalization of key mechanisms and detailed ablation studies showing importance of each component
3. The evaluation is comprehensive as it include a variety of benchmarks testing different aspects of agent heterogeneity along with strong empirical results across all tasks. (GPT3.5-based IoA agents outperforming GPT4 baselines)

**Weaknesses:**

One major concern about this framework is the cost (compute/financial) as Table 6 showed IoA is much more expensive than other multi-agent frameworks like AutoGPT and Open Interpreterm which might limit deployment of this frame at scale.

**Questions:**

How does performance scale with increasing number of agents?

---

> ### Author Response · Authors · 2024-11-20
>
> Thank you for your thoughtful review and for highlighting the novelty, rigorous design, and comprehensive evaluation of our IoA framework. We are especially pleased that you found the architecture well-motivated and the empirical results compelling. Below, we address your concerns.
>
> ---
>
> **1. The Cost of IoA Compared to Other Frameworks**
>
> We completely understand your concern regarding the higher costs associated with IoA compared to other frameworks such as AutoGPT and OpenInterpreter. We have identified three primary factors contributing to this:
>
> 1. **Unoptimized Prompts**: In our experiments, we prioritized clarity and robustness over brevity in prompt design. This has resulted in longer input prompts, increasing the input token cost per LLM call.
> 2. **Repeated Responses**: LLMs occasionally generate redundant responses, repeating prior outputs without contributing new information.
> 3. **Redundant Communication Patterns**: The communication patterns between agents currently mimic human conversational norms, which often include greetings, verbose explanations, and redundant grammar. While this enhances readability, much of this content is unnecessary for task performance and could potentially be omitted or simplified.
>
> To address these cost concerns, we believe there is substantial room for optimization:
>
> - **Optimizing Prompts**: By making prompts more concise and context-aware, we can significantly reduce input token costs, especially since multiple LLM calls occur during task execution. Efforts to streamline prompts are already underway.
> - **Leveraging Advanced Models**: As LLMs like GPT-4 and subsequent iterations improve foundational capabilities and alignment, issues such as repeated responses are expected to diminish, leading to lower output token costs.
> - **Improving Communication Efficiency**: Recent work on enhancing agent communication efficiency, such as sparse communication protocols and compact message representations [1,2,3], can be integrated into IoA. These methods prioritize semantic content over verbose dialogue, substantially reducing token usage.
>
> We believe that IoA serves as an important prototype for LLM-based MAS and that further optimizations will make it more cost-effective while maintaining its performance advantages. We are encouraged by your recognition of our efforts and remain committed to exploring these optimizations in future work.
>
> ---
>
> **2. Scalability with Increasing Number of Agents**
>
> Scalability is a core consideration in IoA’s design. Empirical results from GAIA and RocoBench already demonstrate IoA’s effectiveness with teams of varying sizes. However, the current agent benchmarks rarely require coordination among more than five agents for task-solving, making it challenging totest IoA's performance in larger-scale scenarios.
>
> We believe this reflects a broader challenge in the development of multi-agent benchmarks: the lack of benchmarks that simultaneously require significant collaboration, involve complex problem-solving, and demand scalability to larger agent groups. Existing benchmarks primarily focus on small-scale tasks, which, while valuable, do not fully capture the potential of distributed multi-agent systems like IoA. When suitable and sufficiently complex benchmarks for larger groups of agents are developed, we are eager to extend IoA’s evaluation to those scenarios. In the meantime, our experiments have robustly demonstrated IoA’s effectiveness in not-that-large scale, highlighting its ability to facilitate collaboration and enhance performance. These results provide a strong foundation for scaling IoA to larger teams in future work, and our work may open a new research direction for researchers, providing a possible effective way for scaling MAS.
>
> ---
>
> We hope these responses address your concerns and clarify the cost-related and scalability aspects of IoA. If our explanations alleviate your concerns, we kindly request you to maintain your positive evaluation. Thank you again for your constructive feedback and support, which will guide our future optimizations and research.
>
> ---
>
> **References**
>
> [1] Li, Yunxuan, et al. "Improving Multi-Agent Debate with Sparse Communication Topology." arXiv preprint arXiv:2406.11776 (2024).
>
> [2] Chen, Weize, et al. "Beyond Natural Language: LLMs Leveraging Alternative Formats for Enhanced Reasoning and Communication." arXiv preprint arXiv:2402.18439 (2024).
>
> [3] Chen, Weize, et al. "Optima: Optimizing Effectiveness and Efficiency for LLM-Based Multi-Agent System." arXiv preprint arXiv:2410.08115 (2024).

---

> > ### Comment · Reviewer_78P6 · 2024-11-26
> >
> > Thank you for addressing my concerns. I will stick with my original rating for the paper.

---

### Official Review · Reviewer_3Ebv · 2024-11-04

**Soundness:** 2
**Presentation:** 1
**Contribution:** 3
**Rating:** 6
**Confidence:** 4

**Summary:**

This paper introduces the Internet of Agents framework for multi-agent collaboration, where agents from different developers communicate via an instant-messaging-like architecture and work in teams.
Communication is implemented by abstracting several conversational states, and implementing a finite-state machine to support state transition for agents.
The set of states includes 5 states, representing discussion, synchronous task assignment, asynchronous task assignment, pause & trigger, and conclusion, so and most of the "work" happens in the discussion state (this was not very clear to me).   Each agent has a LLM-based component which determines state transitions and selects the subsequent speaker.
The framework is evaluated using experiments with three types of tasks where teams of agents communicating using Internet of Agents framework are compared to existing multi-agent systems.

**Strengths:**

Inventing a "language" that can enable information transfer between different versions of AI agents is an exciting research question. It is certainly a useful idea, as proliferation of AI agents comes with a proliferation of agent versions and brands. At this stage, the question is still rhetorical rather than practical, as it took decades to build countless networking protocols for wireless routers and phones, like SIP, H323, TCP, UDP,.... In the current work the authors touch upon a novel question of introducing such as language, and take a step toward considering what would this language look like.

**Weaknesses:**

I feel like the authors miss the opportunity to focus the argument as a rhetorical/philosophical consideration to build an analogy with internet communication protocols.
Given the novelty of the issue, the system is still a toy and the experimental evaluation does not look as convincing as the conceptual notion of the potential that such a framework could be.

Part of this general presentation issue, I find the Figure 1 to be much too cluttered. Maybe one way to approach this is to use an open source diagram creation tool, like draw.io, to make a clean arrows and boxes diagram with minimal text, showing how the various components interact.

The paper could benefit from a few task workflow examples.

Overall all these comments are referring to one common point, which is lack of focus and clarity. My impression from this paper is that this is a worthy subject, but it is difficult to assess (rather than infer) what was done.

**Questions:**

How where these three specific benchmarks chosen?

The workflow of the conversational state machine is unclear to me. How are the tasks broken down into components?

Each agent has a LLM component that manages state transitions, and which other components must agents have, to be part of this framework?

Are only LLM-based agents able to participate in the framework?

Is it possible to present results (e.g. Table 3) as a bar plots with confidence intervals? If not, why?

---

> ### Author Response · Authors · 2024-11-20
>
> Thank you for your detailed review and for highlighting the exciting potential of our IoA framework. We appreciate your recognition of the novelty in designing a “language” for agent collaboration and the significant questions this work addresses. Below, we respond to your concerns and clarify key aspects of our work.
>
> ---
>
> **1. IoA only as a Proof of Concept**
>
> We agree that creating a fully functional Internet of Agents is an ambitious and extensive project. In this paper, we present IoA as a proof of concept, focusing on demonstrating its potential to connect heterogeneous and distributed agents. To validate its effectiveness, we evaluated IoA on diverse tasks, including the GAIA benchmark (complex real-world tasks), RocoBench (embodied agent collaboration), and QA with RAG across various datasets. The variety of benchmarks illustrates IoA’s potential in different contexts. While IoA remains a prototype, we believe showing its promise across diverse areas is an important step forward for the field.
>
> ---
>
> **2. Figure 1 Is Too Cluttered**
>
> Thank you for pointing this out. We have redrawn Figure 1 with a simpler and cleaner design using basic arrows and boxes to illustrate the layered architecture and component interactions clearly. The updated figure is accessible via the following anonymous link: https://anonymous.4open.science/r/ioa-rebuttal-606B/architecture.pdf. We will update the submission to include this revised figure, ensuring it enhances the clarity of our presentation. Thank you for your advice.
>
> ---
>
> **3. Example of the Task Workflow**
>
> We agree that providing a concrete example of the workflow would enhance the readers' understanding of the system. While Appendix B includes a brief example, we have now collected a complete interaction trajectory from IoA to demonstrate its workflow in more detail. These trajectories are visualized in new figures, available at https://anonymous.4open.science/r/ioa-rebuttal-606B (files labeled `complete_trajectory_*.pdf`). We will include these figures in the appendix and reference them in the main text to improve clarity and accessibility.
>
> ---
>
> **4. Selection of Benchmarks**
>
> The benchmarks were chosen to cover diverse areas and demonstrate IoA’s versatility:
>
> 1. **GAIA**: A popular, comprehensive benchmark for testing agents on complex real-world tasks, suitable for evaluating IoA’s potential for heterogeneous agent collaboration.
> 2. **RocoBench**: Designed for embodied agents in distributed environments, aligning with IoA’s distributed architecture.
> 3. **QA with RAG**: Reflecting real-world scenarios where distributed agents possess distinct knowledge, requiring collaborative information exchange to improve question answering.
>
> Our goal was to showcase IoA’s effectiveness and potential across diverse task categories, demonstrating its capability to address different collaboration challenges.
>
> ---
>
> **5. Task Breakdown and Workflow of the FSM**
>
> IoA does not explicitly define a separate stage for task decomposition. Instead, as described in Section 2.3.3 and Figure 3, the discussion state is where agents autonomously analyze and decompose tasks into subtasks, leveraging their understanding of teammates’ capabilities. Our experiments show that agents transition from discussion to task assignment stages effectively, with minimal need for external intervention. This behavior highlights the strength of IoA’s flexible design, and shows that current LLMs are capable of adapting their behavior to our IoA design.
>
> ---
>
> **6. Components Required for Participation in IoA**
>
> We use one LLM to handle most of the framework’s mechanisms, including state transition, task assignment, summarizing execution results, etc. Our experiments demonstrate that SoTA LLMs such as GPT-4 and GPT-3.5-turbo, even with simple prompts, are effective in IoA. Key required abilities for an LLM in IoA may include:
>
> 1. Task decomposition and assignment based on teammate capabilities.
> 2. Understanding IoA’s framework and mechanisms for state transition.
> 3. Correctly interpreting and summarizing tasks and results.
>
> As LLM technology advances, we anticipate broader accessibility and higher performance for IoA-compatible agents.

---

> > ### Author Response · Authors · 2024-11-20
> >
> > **7. Non-LLM Agents in IoA**
> >
> > Currently, IoA focuses on LLM-based agents due to their conversational and reasoning capabilities. However, non-LLM-based agents can participate by functioning as tools within LLM-based agents. For example, a non-LLM agent can expose APIs that an LLM calls during task execution. To support non-LLM agents to directly operate within IoA, the protocol may need to be updated. We leave it to future work, and focus mainly on showing the effectiveness of connecting LLM-based agents.
> >
> > ---
> >
> > **8. Can Table 3 be presented in Bar Plots with Confidence Intervals**
> >
> > We appreciate your suggestion and agree that bar plots with confidence intervals is an alternative presentation. However, as baseline results in [1] are reported as single-point values without confidence intervals, we followed the same convention to maintain consistency. We are confident in the robustness of our results, as the performance gaps (e.g., IoA’s 0.671 versus baselines using GPT-3.5-turbo) are substantial and unlikely to be attributed to noise.
> >
> > ---
> >
> > We hope these responses address your concerns and clarify the scope and strengths of IoA. If our explanations have alleviated your concerns, we kindly request you to consider revising your evaluation. Thank you for your thoughtful and constructive feedback, which will greatly enhance the clarity and impact of our work.
> >
> > [1] Wang, Haotian, et al. "Apollo's Oracle: Retrieval-Augmented Reasoning in Multi-Agent Debates." *arXiv preprint arXiv:2312.04854* (2023).

---

> > > ### Comment · Reviewer_3Ebv · 2024-12-01
> > > **Thank you for your detailed responce.**
> > >
> > > I have read the authors response, and I am increasing my rating up to a 6.

---

### Official Review · Reviewer_kxKm · 2024-11-09

**Soundness:** 3
**Presentation:** 4
**Contribution:** 4
**Rating:** 8
**Confidence:** 3

**Summary:**

This paper introduces the Internet of Agents (IoA), a novel framework for LLM-based multi-agent collaboration, inspired by the concept of the Internet. The framework aims to facilitate collaboration among autonomous agents by addressing the limitations of existing multi-agent frameworks, including ecosystem isolation, single-device simulation, and rigid communication and coordination. IoA consists of two core components: the server and the client. The server acts as a central hub controller, while the client serves as a wrapper for individual agents, enabling the necessary communications and protocols. The IoA architecture has three layers, i.e., interaction, data, and foundation layers, they collectively accelerate agent collaboration through the network. Experiments conducted in various settings demonstrate the effectiveness and versatility of IoA in facilitating efficient collaboration among heterogeneous agents.

**Strengths:**

The paper is well-written and organized, with the key components of IoA clearly illustrated and easy to follow.

The experiments are comprehensive, assessing IoA's capability to integrate agents with heterogeneous tools on the GAIA benchmark. It introduces a robust benchmark comprising 153 open-ended instructions spanning four diverse categories. The results illustrate IoA's effectiveness in integrating, orchestrating, and enabling independently developed agents with heterogeneous architectures. Further experiments with RoCoBench highlight IoA's ability to enhance the communication and collaboration capabilities of embodied agents.
Moreover, in the RAG question-answering domain, IoA also surpasses existing methods.

**Weaknesses:**

In Section 3.2, while the paper constructs 153 open-ended instructions spanning four categories—search & report, coding, math, and life assistant—there is a lack of sufficient or list of examples (only one figure) to illustrate whether the questions are complex and require multi-step reasoning, or if they can be directly solved.  Therefore, the authors could include an appendix with a representative sample of tasks & solutions from each category, or provide a link to the full benchmark.

Additionally, the paper primarily employs GPT-4 as an impartial judge of response quality, but IoA might require more steps than competitors such as AutoGen.

**Questions:**

It would be beneficial to provide additional examples related to the constructed benchmark and to illustrate the specific steps undertaken by IoA.

GPT-4 might occasionally give higher evaluation scores or exhibit overconfidence. Could you discuss any potential biases or limitations associated with using this evaluation method in your benchmarks and whether you considered alternative evaluation approaches.

---

> ### Author Response · Authors · 2024-11-20
>
> Thank you for your thoughtful and detailed review, and for recognizing the clarity, organization, and contributions of our work. We are delighted that you found the IoA framework, experimental design, and results compelling. Below, we address your comments and suggestions in detail.
>
> ---
>
> **1. Lack of Representative Examples in the Constructed Benchmark**
>
> The figures are available at the anonymous link: https://anonymous.4open.science/r/ioa-rebuttal-606B. Files labeled `responses_*.pdf` include representative instructions from the open-ended instruction benchmark and the corresponding responses from various frameworks. Additionally, files labeled `complete_trajectory_*.pdf` present complete communication trajectories from GAIA and the open-ended instruction benchmark, offering a detailed view of IoA's workflow and performance. We hope these additions address your concerns effectively.
>
> ---
>
> **2. Discussion on Potential Biases in Using GPT-4 for Evaluation**
>
> We acknowledge the potential biases and limitations associated with using GPT-4 as an evaluation metric. While GPT-4 has demonstrated strong alignment with human judgment in prior works [1], overconfidence and subtle biases can occasionally affect its scoring. To mitigate these issues:
>
> 1. **Robust Evaluation Protocols**: We adopt the protocol from [1], alternating the order of responses during evaluation to minimize order-induced biases.
> 2. **Bias Mitigation through Diversity**: The instructions in our benchmark span diverse categories, reducing the potential for category-specific biases in GPT-4’s scoring.
>
> While we considered alternative approaches, such as human evaluation or using multiple LLM judges, these introduce scalability challenges. GPT-4 remains a practical choice for large-scale evaluation, and its agreement with human evaluators in prior works supports its reliability. Also, it is widely adopted by current LLM judgment, e.g., MT-Bench [1], Arena-Hard [2], Alpaca-Eval [3,4].
>
> We will add a discussion in the appendix explicitly outlining the potential limitations of GPT-4 in the evaluation and our strategies to address them.
>
> ---
>
> We hope these responses address your concerns and further enhance the clarity and rigor of our work. If these clarifications alleviate your concerns, we kindly request you to consider maintaining your positive evaluation. Thank you again for your constructive feedback and insightful questions.
>
> ---
>
> [1] Zheng, L., et al. "Judging LLM-as-a-judge with MT-Bench and Chatbot Arena." Advances in Neural Information Processing Systems 36 (2023): 46595-46623.
>
> [2] Li, Tianle, et al. "From Crowdsourced Data to High-Quality Benchmarks: Arena-Hard and BenchBuilder Pipeline." *arXiv preprint arXiv:2406.11939* (2024).
>
> [3] Li, Xuechen, et al. "Alpacaeval: An automatic evaluator of instruction-following models." 2023,
>
> [4] Dubois, Yann, et al. "Length-controlled alpacaeval: A simple way to debias automatic evaluators." *arXiv preprint arXiv:2404.04475* (2024).

---

### Official Review · Reviewer_gNTA · 2024-11-09

**Soundness:** 3
**Presentation:** 3
**Contribution:** 3
**Rating:** 6
**Confidence:** 4

**Summary:**

This paper proposes the "Internet of Agents" (IoA) framework to facilitate enhanced collaboration among autonomous agents using large language models (LLMs). Inspired by the Internet’s structure, IoA aims to address the integration challenges seen in current multi-agent systems (MAS). This is achieved through three key innovations: a protocol for integrating diverse agents, a layered architecture resembling an instant-messaging framework, and mechanisms for dynamic team formation and conversation flow. Through a series of experiments, the paper claims that IoA enables more effective collaboration than existing MAS frameworks, particularly in domains like retrieval-augmented generation (RAG) and embodied AI tasks.

**Strengths:**

- Innovative Conceptual Framework: The IoA draws a compelling parallel between the functionality of MAS and the Internet, introducing layered architecture and team formation protocols that offer adaptability and scalability.

- Comprehensive Experiments: The paper provides a well-rounded experimental design, including benchmarks that demonstrate the system’s effectiveness across multiple task categories.

- Task-Specific Improvements: IoA demonstrates improved performance in benchmarks like the GAIA and RAG, highlighting its potential advantage over existing MAS in facilitating agent cooperation and integrating heterogeneous tools.

**Weaknesses:**

- Lack of Detailed Evaluation on Real-World Applications: While the experimental benchmarks are extensive, the paper lacks real-world application scenarios that demonstrate the IoA’s effectiveness in real world environments that require distributed settings. It is not unclear why GAIA benchmarks would need agents over internet instead of a simple central multi-agent system.

- Scalability and Performance Analysis: The paper lacks analysis on IoA’s computational overhead and network latency issues, especially when scaled to large numbers of agents across different devices distributed settings. An understanding of these scalability and performance trade-offs is critical.

- Vagueness in Mechanisms for Conversation Flow Control: The conversation flow control relies on LLMs and a finite state machine (FSM) for transitions, but the scalability of this approach when coordinating many situations with many states is not well justified. It would be great to have some ablation studies.

- Limited Insight into Integration of Third-Party Agents: The implementation of heterogeneous agent integration is described conceptually, but details on handling diverse architectures and security considerations when integrating third-party agents are lacking.

- Limited Novel Research Contributions: The paper’s focus on architecture and engineering limits its contribution to ML research. While the framework offers novel infrastructure, the lack of new research methodologies or performance optimizations in agent interactions might reduce its appeal in ML-centric venues.

**Questions:**

- How does IoA handle latency and data synchronization when agents are distributed across devices in real time?

- How does the protocol for agent integration address security, especially for third-party agents that may have differing trust levels? Could the authors expand on IoA’s security measures, especially regarding agent registration and message routing?

- Can the FSM for conversation flow control be adapted or scaled effectively if used with a larger number of agents than tested in the experiments? How adaptable is the finite-state machine-based flow control to tasks requiring high real-time interaction or rapid state changes?

---

> ### Author Response · Authors · 2024-11-20
>
> Thank you for your thoughtful review and for recognizing the innovative aspects of our framework. We are delighted that you found the strengths of IoA compelling, and we appreciate your detailed feedback, which provides valuable directions for enhancing our work. Below, we address each of your comments in detail.
>
> ---
>
> **1. Lack of Detailed Evaluation on Real-World Applications**
>
> Thank you for raising this important point. We completely agree that real-world application evaluations would further strengthen our claims. But we have to say current benchmarks for multi-agent systems lack the complexity and metrics suitable for benchmarking of distributed, heterogeneous agents. Therefore, in our paper, we utilized GAIA and embodied task benchmarks like RocoBench, which simulate real-world conditions requiring agent cooperation and diverse capabilities. While GAIA might not intrinsically require distributed agents, it effectively tests coordination and problem-solving capabilities in complex tasks.
>
> Actually, we have conducted experiments deploying IoA to smart home devices, allowing agents to manage daily routines across distributed devices (e.g., adjusting lighting via a home assistant based on notifications from a portable device agent). Although these experiments show promising results, they are challenging to quantize and standardize for publication, we choose to not present them in our paper. Constructing a real-world benchmark of multi-agent is challenging, and constructing such benchmark could itself be a solid work. We temporarily leave it for future work.
>
> ---
>
> **2. Scalability and Performance Analysis**
>
> Scalability is indeed critical for IoA. As highlighted in Appendix E (Table 6), we report costs, which partially reflect IoA's computational overhead as it is calculated with the number of input tokens and output tokens. Directly quantifying network latency in distributed settings is complex due to varying device configurations and network conditions. Nonetheless, our architectural design tried to enhance the scalability through features like nested team formation, which reduces communication overhead by segmenting tasks into smaller, manageable group chats. We will mention the computational overhead and the designs that tried to address the scalability concerns explicitly in the main text.
>
> ---
>
> **3. Scalability of FSM**
>
> We address your concerns of FSM proposed in the weakness and question here. The FSM is designed to operate efficiently within small group chats. For scenarios with a large number of agents, our nested team formation mechanism creates hierarchical subgroups. This minimizes the complexity and ensures FSM efficiency within each subgroup. Recent work has also shown the effectiveness and efficiency of such sparse communication topology [1]. We will expand on this mechanism in the paper to clarify how scalability is achieved through decentralized team formation.
>
> ---
>
> **4. Limited Insight into Integration of Third-Party Agents**
>
> We appreciate this observation and will add implementation details to the appendix for transparency. These are mostly engineering work so we did not include them in the initial submission. Basically, each third-party agent is wrapped as a Docker container with an adapter that unifies its input/output interfaces, facilitating seamless integration. For example, AutoGPT's integration involved invoking its `steps` method repeatedly when implementing the `run` API of its adapter. These adapters handle agent-specific design while presenting a consistent interface to the IoA framework. We will also ensure this process is well-commented in our open-source repository upon its release.
>
> ---
>
> **5. Limited Novel Research Contributions**
>
> While IoA emphasizes architecture and engineering, we believe it addresses a fundamental gap in LLM-based MAS by enabling heterogeneous, distributed agent collaboration. This aligns with the ICLR track that we submit to, which focuses on "applications to language and other modalities." Furthermore, our extensive benchmarks and novel features, such as the layered architecture and dynamic team formation, establish IoA's practical value. We argue that “agent” is itself a rather empirical research area, and the effectiveness can sometimes be important, as agent should be “useful”. The presented results underscore IoA’s potential for advancing both empirical research and real-world deployment.

---

> > ### Author Response · Authors · 2024-11-20
> >
> > **6. Latency and Data Synchronization**
> >
> > Currently, IoA follows an instant messaging model where agents communicate through a central server. Data synchronization occurs when agents exchange updates upon receiving new information. While this approach is not robust to significant network fluctuations, it provides a practical proof of concept for effective distributed agent communication. Addressing latency and synchronization comprehensively is an engineering-intensive challenge that we aim to explore in future work, and right now, we think that proving the research value of such direction is more important.
> >
> > ---
> >
> > **7. Security Concerns**
> >
> > Security is a crucial area for future research. IoA introduces a realistic scenario for investigating multi-agent security, such as secure agent registration and message routing. We acknowledge that addressing comprehensive security measures requires significant engineering efforts and plan to engage with this broader topic in subsequent work.
> >
> > ---
> >
> > We hope these responses address your concerns and demonstrate our commitment to refining IoA for broader applicability. If any of these clarifications alleviate your concerns, we kindly request you to consider raising your evaluation. Thank you for your constructive feedback and valuable insights.
> >
> > [1] Li, Yunxuan, et al. "Improving Multi-Agent Debate with Sparse Communication Topology." *arXiv preprint arXiv:2406.11776* (2024).

---

### Meta-Review · Area_Chair_oFf1 · 2024-12-19

**Metareview:**

By drawing inspiration from the concept of the Internet, this paper proposes an LLM-based multi-agent collaboration framework, Internet of Agents (IoA).  Experimental results have demonstrated that IoA can enhance multi-agent collaboration. This work proposed a novel, technically solid framework for multi-agent systems. All reviews are positive and the authors have addressed reviewers' concerns.

**Additional Comments On Reviewer Discussion:**

There was only one initial review that was marginally below acceptance before rebuttal. The reviewer was satisfied with the rebuttal and raised the score to above acceptance. The final recommendations were unanimous.

---

### Decision · Program_Chairs · 2025-01-22

Accept (Spotlight)